# UltraTWD: Optimizing Ultrametric Trees for Tree-Wasserstein Distance

**Fangchen Yu** [1,2]  **Yanzhen Chen** [1]  **Jiaxing Wei** [1]  **Jianfeng Mao** [1,3]  **Wenye Li** [1,4]*  **Qiang Sun** [2,5]*

## Abstract

The Wasserstein distance is a widely used metric for measuring differences between distributions, but its super-cubic time complexity introduces substantial computational burdens. To mitigate this, the tree-Wasserstein distance (TWD) offers a linear-time approximation by leveraging a tree structure; however, existing TWD methods often compromise accuracy due to suboptimal tree structures and edge weights. To address it, we introduce UltraTWD, a novel unsupervised framework that simultaneously optimizes both ultrametric tree structures and edge weights to more faithfully approximate the cost matrix. Specifically, we develop algorithms based on minimum spanning trees, iterative projection, and gradient descent to efficiently learn high-quality ultrametric trees. Empirical results across document retrieval, ranking, and classification tasks demonstrate that UltraTWD achieves superior approximation accuracy and competitive downstream performance. Code is available at: https://github.com/NeXAIS/UltraTWD.

## 1. Introduction

The Wasserstein distance (Villani, 2008) is a powerful metric for measuring dissimilarities between probability distributions, with various applications (Takezawa et al., 2021; Khrulkov et al., 2023) A well-known example, the Word Mover's Distance (Kusner et al., 2015), computes distances between documents modeled as bag-of-words distributions over $n$ word embeddings. Despite its effectiveness, the Wasserstein distance suffers from a high computational complexity of $\mathcal{O}(n^3 \log n)$, limiting its practical applicability.

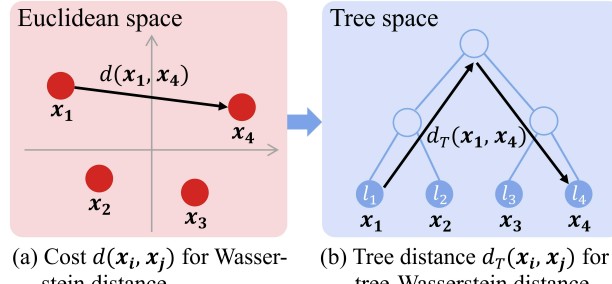

Euclidean space  Tree space

$d(\boldsymbol{x_1}, \boldsymbol{x_4})$

(a) Cost $d(\boldsymbol{x_i}, \boldsymbol{x_j})$ for Wasserstein distance

(b) Tree distance $d_T(\boldsymbol{x_i}, \boldsymbol{x_j})$ for tree-Wasserstein distance

*Figure 1.* The tree-Wasserstein distance embeds data points into a tree and replaces the cost $d(\boldsymbol{x_i}, \boldsymbol{x_j})$ with tree distance $d_T(\boldsymbol{x_i}, \boldsymbol{x_j})$.

Several approximation methods have been proposed to mitigate the computational challenges of the Wasserstein distance. The Sinkhorn distance (Cuturi, 2013) reduces the complexity to $\mathcal{O}(n^2)$ by introducing an entropic regularization term. The sliced-Wasserstein distance (SWD) (Rabin et al., 2012) achieves $\mathcal{O}(n \log n)$ complexity by projecting high-dimensional data onto one-dimensional spaces and computing distances over these 1D projections. The tree-Wasserstein distance (TWD) (Le et al., 2019) further improves efficiency by learning a rooted tree, enabling $\mathcal{O}(n)$ complexity, as illustrated in Fig. 1. This work focuses on advancing the TWD methodology.

Existing TWD methods can be categorized into four categories, each with notable limitations. **(1) Tree-construction methods:** QuadTree (Indyk & Thaper, 2003) and ClusterTree (Le et al., 2019) construct trees by partitioning hypercubes through recursive division or farthest-point clustering (Gonzalez, 1985). However, their edge weights, determined by the depth of nodes, are often suboptimal. **(2) Weight-optimized methods:** qTWD and cTWD (Yamada et al., 2022) refine the edge weights of QuadTree or ClusterTree using Lasso regression but fail to optimize the tree structures themselves, resulting in performance heavily dependent on the randomness of the initial tree. **(3) Tree-sliced methods:** Sliced-qTWD and cTWD (Yamada et al., 2022; Otao & Yamada, 2023) improve stability by averaging multiple random trees, yet they cannot optimize tree structures. **(4) Supervised method:** UltraTree (Chen et al., 2024) learns an ultrametric tree by regressing TWD against the true Wasserstein distance. However, this approach requires extensive training data with precomputed Wasserstein distances as labels, which makes it computationally expensive.

[1]The Chinese University of Hong Kong, Shenzhen [2]Mohamed bin Zayed University of Artificial Intelligence [3]Shenzhen Research Institute of Big Data [4]The Hong Kong University of Science and Technology (Guangzhou) [5]University of Toronto. Correspondence to: Wenye Li <wenyeli@hkust-gz.edu.cn>, Qiang Sun <qsun-stats@gmail.com>.

*Proceedings of the 42nd International Conference on Machine Learning*, Vancouver, Canada. PMLR 267, 2025. Copyright 2025 by the author(s).

Given these challenges, unsupervised methods fail to optimize tree structures and are susceptible to random initialization, while the supervised method depends on expensive labeled training data. These limitations motivate us to develop a new framework that *optimizes both tree structure and edge weights without requiring training data*. To achieve this, we propose UltraTWD, a novel unsupervised framework that simultaneously learns the ultrametric tree structure and edge weights, delivering enhanced accuracy. To ensure the tree distance $d_T$ closely approximates the cost $d$, our method constructs an ultrametric tree equipped with an ultrametric $D_T = [d_T(\boldsymbol{x_i}, \boldsymbol{x_j})] \in \mathbb{R}^{n \times n}$ that aligns with the cost matrix $D = [d(\boldsymbol{x_i}, \boldsymbol{x_j})] \in \mathbb{R}^{n \times n}$. Specifically, this is achieved by solving **ultrametric nearness problems**, such as

$$\min_{D_T} \|D_T - D\|_\infty \quad \text{or} \quad \min_{D_T} \|D_T - D\|_F^2.$$

Throughout the optimization, both the tree structure and edge weights are iteratively refined, resulting in a more accurate and robust tree-Wasserstein distance.

Our contributions are summarized as follows:

• We introduce UltraTWD, a new framework for tree-Wasserstein distance that leverages the ultrametric property. To our knowledge, this is the first unsupervised framework to simultaneously optimize tree structure and edge weights, overcoming the reliance of prior work on fixed tree structures, multiple trees, or extensive training data.

• We formulate ultrametric nearness problems to construct ultrametric trees that closely approximate a given cost matrix. To solve these problems efficiently, we propose three algorithms: (1) a fast minimum spanning tree-based approach, (2) an iterative projection method, and (3) a gradient descent-based method, all yielding high-quality solutions.

• Our UltraTWD achieves the lowest estimation errors among unsupervised methods and even surpasses the supervised method across four benchmark datasets. Furthermore, our framework demonstrates strong empirical performance in document retrieval, ranking, and classification tasks, highlighting its practicality for Wasserstein-based applications.

## 2. Preliminaries

### 2.1. 1-Wasserstein Distance

The Wasserstein distance quantifies the difference between distributions. In text analysis, each word is represented by an embedding vector $\boldsymbol{x_i} \in \mathbb{R}^d$, with the vocabulary $X = [\boldsymbol{x_1}, \ldots, \boldsymbol{x_n}] \in \mathbb{R}^{d \times n}$ containing $n$ words. A document $\mu$ is modeled as a normalized bag-of-words distribution, expressed as $\sum_{i=1}^n a_i \delta_{\boldsymbol{x_i}}$, where $\delta_{\boldsymbol{x_i}}$ is the Dirac delta at $\boldsymbol{x_i}$, mass $a_i$ represents the frequency of the $i$-th word in $\mu$, and $\sum_{i=1}^n a_i = 1$. In this paper, the Wasserstein distance refers to the 1-Wasserstein distance, defined as follows:

**Definition 1** (**1-Wasserstein Distance**). *Given two distributions $\mu = \sum_{i=1}^n a_i \delta_{\boldsymbol{x_i}}$ and $\nu = \sum_{j=1}^n b_j \delta_{\boldsymbol{x_j}}$, the 1-Wasserstein distance between $\mu$ and $\nu$ is defined as:*

$$W_1(\mu, \nu) := \min_{\gamma \in \Gamma(\mu,\nu)} \sum_{i,j=1}^n \gamma_{ij} d(\boldsymbol{x_i}, \boldsymbol{x_j}), \quad (1)$$

*where the cost $d(\boldsymbol{x_i}, \boldsymbol{x_j})$ represents the distance between $\boldsymbol{x_i}$ and $\boldsymbol{x_j}$, and $\gamma$ is the transport plan in $\Gamma(\mu, \nu)$ defined as*

$$\Gamma(\mu, \nu) = \{\gamma \in \mathbb{R}_+^{n \times n} \mid \sum_j \gamma_{ij} = a_i, \sum_i \gamma_{ij} = b_j\}, \quad (2)$$

*where $\boldsymbol{a}^\top = [a_1, \ldots, a_n]$ and $\boldsymbol{b}^\top = [b_1, \ldots, b_n]$ are mass vectors satisfying the total mass $\sum_{i=1}^n a_i = \sum_{j=1}^n b_j = 1$.*

Eq. (1) minimizes the total cost of transforming distribution $\mu$ into $\nu$, solvable using a linear programming algorithm with a computational complexity of $\mathcal{O}(n^3 \log n)$ (Villani, 2008). With Euclidean distance as the cost, $W_1$ corresponds to the Word Mover's Distance (Kusner et al., 2015).

### 2.2. Tree-Wasserstein Distance

The tree-Wasserstein distance (TWD) efficiently approximates the 1-Wasserstein distance using a tree structure. Given the vocabulary $X = [\boldsymbol{x_1}, \boldsymbol{x_2}, \ldots, \boldsymbol{x_n}] \in \mathbb{R}^{d \times n}$, each word vector $\boldsymbol{x_i}$ is embedded into a corresponding leaf node $l_i$ in the tree (Fig. 2). This tree structure enables TWD to provide a closed-form solution and achieve efficient computation with a linear complexity of $\mathcal{O}(n)$, where $n$ is both the vocabulary size and number of leaves.

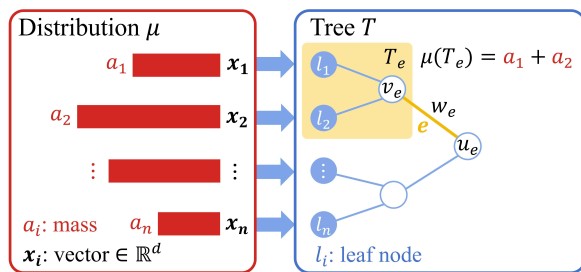

*Figure 2.* Illustration of the tree-Wasserstein distance.

**Definition 2** (**Tree-Wasserstein Distance**). *Given a tree $T$ with tree distance $d_T$, the tree-Wasserstein distance between distributions $\mu$ and $\nu$ is defined as (Yamada et al., 2022):*

$$W_T(\mu, \nu) := \min_{\gamma \in \Gamma(\mu,\nu)} \sum_{i,j=1}^n \gamma_{ij} d_T(\boldsymbol{x_i}, \boldsymbol{x_j}), \quad (3)$$

*where $d_T(\boldsymbol{x_i}, \boldsymbol{x_j})$ represents the shortest path distance between leaves $l_i$ and $l_j$, and $\gamma$ is the transport plan as defined in Eq. (2). The TWD has the following analytical form:*

$$W_T(\mu, \nu) = \sum_{e \in T} w_e \cdot |\mu(T_e) - \nu(T_e)|, \quad (4)$$

*where $w_e$ denotes the weight of edge $e \in T$, and $\mu(T_e)$, $\nu(T_e)$ are the total masses within the subtree $T_e$ rooted at the deeper node of edge $e$, as illustrated in Fig. 2.*

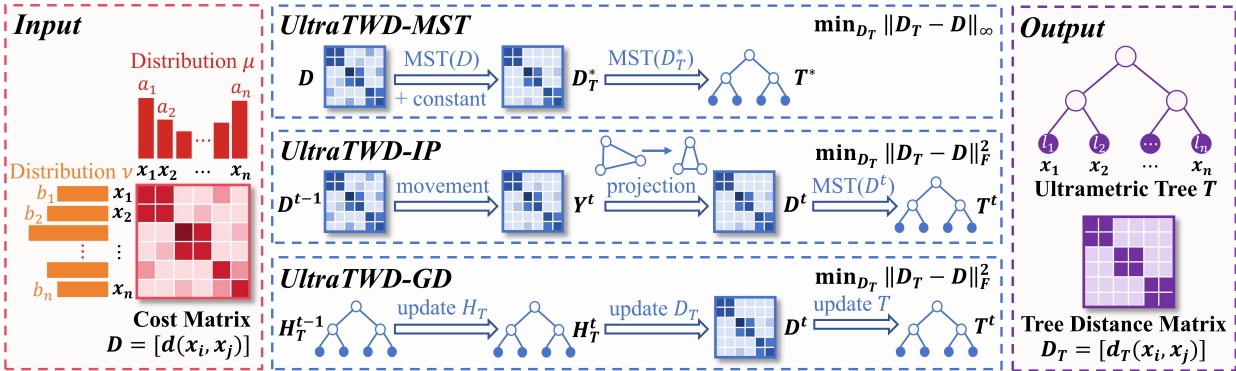

*Figure 3.* A diagram of the proposed UltraTWD framework for tree-Wasserstein distance computation. UltraTWD-MST applies minimum spanning trees (MST) to minimize $\|D_T - D\|_\infty$ with an optimal guarantee. To minimize $\|D_T - D\|_F^2$, UltraTWD-IP employs iterative projection (IP) on the distance matrix, while UltraTWD-GD uses gradient descent (GD) for efficient tree structure optimization.

# 3. Unsupervised Tree-Wasserstein Distance Based on Ultrametric Trees

In this section, we first analyze the approximation gap between $W_T$ and $W_1$, motivating our formulation of two ultrametric nearness problems. We then present the UltraTWD framework with three unsupervised approaches for ultrametric optimization, concluding with the algorithm analysis.

## 3.1. Bridging the Gap between $W_T$ and $W_1$

The discrepancy between the tree-Wasserstein distance $W_T$ and the 1-Wasserstein distance $W_1$ primarily stems from the difference between the tree distance $d_T$ and the cost $d$. As defined in Eqs. (1) and (3), if $d_T(\boldsymbol{x_i}, \boldsymbol{x_j}) = d(\boldsymbol{x_i}, \boldsymbol{x_j})$ for all pairs $(\boldsymbol{x_i}, \boldsymbol{x_j})$, then $W_T(\mu, \nu) \equiv W_1(\mu, \nu)$ for any distributions $\mu$ and $\nu$ (Yamada et al., 2022). However, this ideal scenario is rarely achievable because the tree distance matrix $D_T = [d_T(\boldsymbol{x_i}, \boldsymbol{x_j})] \in \mathbb{R}^{n \times n}$ must be a tree metric, whereas the cost matrix $D = [d(\boldsymbol{x_i}, \boldsymbol{x_j})] \in \mathbb{R}^{n \times n}$ generally does **NOT** conform to a tree metric.

**Definition 3** (**Distance and Tree Metric** (Semple & Steel, 2003)). *Consider a symmetric, non-negative matrix $D \in \mathbb{R}^{n \times n}$ with zero diagonal values. For all $1 \le i, j, k, l \le n$:*
• *$D$ is a **distance metric** if it satisfies the **triangle inequality**:*

$$d_{ij} \le d_{ik} + d_{jk}; \qquad (5)$$

• *$D$ is a **tree metric** if it satisfies the **four-point condition**:*

$$d_{ij} + d_{kl} \le \max\{d_{ik} + d_{jl}, \ d_{il} + d_{jk}\}. \qquad (6)$$

From Definition 3, a tree metric must be a distance metric, as setting $l = k$ in Eq. (6) implies Eq. (5). However, a distance metric is not necessarily a tree metric. For example, the Euclidean distance matrix $D = \begin{bmatrix} 0 & 3 & 5 & 4 \\ 3 & 0 & 4 & 5 \\ 5 & 4 & 0 & 3 \\ 4 & 5 & 3 & 0 \end{bmatrix}$ satisfies all triangle inequalities but fails the four-point condition, due to $d_{13} + d_{24} = 10 > 8 = \max\{d_{12} + d_{34}, \ d_{14} + d_{23}\}$.

The fundamental mismatch between $D_T$ and $D$ creates a gap between $W_T$ and $W_1$. Bridging this gap requires finding a tree metric $D_T$ that closely approximates the given cost matrix $D$, which serves as the motivation for our framework.

## 3.2. Formulating Ultrametric Nearness Problems

To closely approximate $D$, we first formulate the **tree-metric nearness problem**, which aims to find the nearest tree metric $D_T$ to the given cost matrix $D \in \mathbb{R}^{n \times n}$:

$$\min_{D_T \in \mathbb{R}^{n \times n}} \|D_T - D\|,$$
$$\text{s.t.} \quad d_{ij}^T + d_{kl}^T \le \max\{d_{ik}^T + d_{jl}^T, \ d_{il}^T + d_{jk}^T\}, \qquad (7)$$
$$d_{ii}^T = 0, d_{ij}^T = d_{ji}^T \ge 0, \forall \, 1 \le i, j, k, l \le n,$$

where $d_{ij}^T$ represents $d_T(\boldsymbol{x_i}, \boldsymbol{x_j})$. However, solving this problem is intractable due to its $\mathcal{O}(n^4)$ constraints involving four points and the non-convex nature of the optimization.

To simplify the problem, we focus on ultrametrics, a special subset of tree metrics. In a rooted tree $T$, the tree distance $d_T(\boldsymbol{x_i}, \boldsymbol{x_j})$ between two leaves $l_i$ and $l_j$ is defined as the height of their least common ancestor (LCA):

$$d_T(\boldsymbol{x_i}, \boldsymbol{x_j}) := h(\text{LCA}(l_i, l_j)). \qquad (8)$$

This definition induces an ultrametric $D_T = [d_T(\boldsymbol{x_i}, \boldsymbol{x_j})] \in \mathbb{R}^{n \times n}$, and $T$ is referred to as an ultrametric tree.

**Definition 4** (**Ultrametric** (Semple & Steel, 2003)). *A tree metric $D_T \in \mathbb{R}^{n \times n}$ is an **ultrametric** if and only if it satisfies the **strong triangle inequality**:*

$$d_{ij}^T \le \max\{d_{ik}^T, \ d_{jk}^T\}, \quad \forall \, 1 \le i, j, k \le n. \qquad (9)$$

For an ultrametric tree, the edge weight is defined as $w_e = \frac{1}{2}\big(h(u_e) - h(v_e)\big)$, where $u_e$ ($v_e$) is the parent (child) node; see Fig. 2. Then, $d_T(\boldsymbol{x_i}, \boldsymbol{x_j})$ in Eq. (8) equals the shortest path distance between $l_i$ and $l_j$, consistent with Definition 2. See Appendix A.3 for ultrametric background.

Focusing on ultrametrics simplifies Problem (7) into the following **ultrametric nearness problems**, which aim to find the nearest ultrametric $D_T$ to a given cost matrix $D$. Specifically, we consider two optimization objectives:

$$\min_{D_T \in \mathbb{R}^{n \times n}} \|D_T - D\|_\infty, \qquad (10)$$

or,

$$\min_{D_T \in \mathbb{R}^{n \times n}} \|D_T - D\|_F^2, \qquad (11)$$

subject to the constraints:

$$d_{ij}^T \leq \max\{d_{ik}^T, d_{jk}^T\}, d_{ii}^T = 0, d_{ij}^T = d_{ji}^T \geq 0, \forall i, j, k \in [n].$$

• **The infinity norm** is defined as $\|X\|_\infty := \max_{i,j} |X_{ij}|$ and minimizes the largest entrywise deviation between $D_T$ and $D$. By controlling the maximum error, this objective ensures that $D_T$ avoids significant outliers.

• **The Frobenius norm** is defined as $\|X\|_F^2 := \sum_{i,j} X_{ij}^2$ and minimizes the overall squared error between $D_T$ and $D$. This objective distributes the error more evenly across all entries, resulting in a good average approximation.

As illustrated in Fig. 3, we propose the UltraTWD framework to minimize these two objectives, introducing three approaches for optimizing ultrametrics, detailed as follows.

### 3.3. Optimizing Ultrametrics with Infinity Norm

To compute the $l_\infty$-nearest ultrametric of a cost matrix $D$, we first construct its minimum spanning tree (**MST**) using Prim's algorithm (Prim, 1957), which yields an ultrametric tree $T$ and the corresponding ultrametric $D_T$ (Chen et al., 2024). For simplicity, we collectively denote both $T$ and $D_T$ as MST($D$). We then refine $D_T$ to obtain the $l_\infty$-nearest ultrametric $D_T^*$ using Theorem 5.

**Theorem 5** ($l_\infty$-**Nearest Ultrametric** (Chepoi & Fichet, 2000))**.** *For a distance metric $D \in \mathbb{R}^{n \times n}$, an optimal ultrametric $D_T^*$ that minimizes $\|D_T - D\|_\infty$ is given by:*

$$D_T^* = MST(D) + \frac{1}{2}\|MST(D) - D\|_\infty \mathbf{1}, \qquad (12)$$

*where $\mathbf{1}_{ij} = 1$ if $i \neq j$, and $0$ otherwise, for all $i, j \in [n]$.*

The algorithm is summarized below. Since $D_T^*$ is an ultrametric, the corresponding ultrametric tree $T^*$ can be directly constructed using the MST. While prior works (Chen et al., 2024; Lin et al., 2025) have used or compared against the MST (Prim, 1957), to the best of our knowledge, we are the first to leverage the $l_\infty$-nearest ultrametric for computing the tree-Wasserstein distance.

---

**Algorithm 1 UltraTWD-MST (Minimum Spanning Tree)**

**Input:** $D \in \mathbb{R}^{n \times n}$: cost matrix.
**Output:** $T^*$: optimal ultrametric tree under infinity norm.
 1: Compute the $l_\infty$-nearest ultrametric via Eq. (12):

$$D_T^* = MST(D) + \frac{1}{2}\|MST(D) - D\|_\infty \mathbf{1}.$$

 2: Construct the ultrametric tree: $T^* = MST(D_T^*)$.

---

### 3.4. Optimizing Ultrametrics with Frobenius Norm: Iterative Projection Method

Unlike the $l_\infty$-nearest ultrametric, which has a closed-form solution, the Frobenius-norm-based ultrametric nearness problem (Eq. (11)) is NP-hard (Křivánek, 1988). Prior methods either relax the strong triangle inequalities into unconstrained objectives with cluster-based regularization (Chierchia & Perret, 2019; Cohen-Addad et al., 2018; Chatziafratis et al., 2018), limiting their use to hierarchical clustering, or focus on alternative triplet relations like LCA relations (Wang & Wang, 2020; Lin et al., 2025), which do not guarantee the ultrametric property. In contrast, we directly enforce the triplet constraint $d_{ij}^T \leq \max\{d_{ik}^T, d_{jk}^T\}$ and solve Eq. (11) through a matrix optimization method.

Although exact minimization is extremely challenging, we propose an approximate solution using **iterative projections** (**IP**). The key idea is to iteratively enforce the ultrametric constraints for each triplet $(i, j, k)$.

First, we solve the sub-problem $\min_{D_T \in \Omega_{ijk}} \|D_T - D\|_F^2$, where the subset $\Omega_{ijk}$ imposes constraints for a single triplet:

$$\Omega_{ijk} := \{D_T \in \mathbb{S}^n \mid d_{ij}^T \leq \max\{d_{ik}^T, d_{jk}^T\}\},$$

with $\mathbb{S}^n$ representing the set of $n \times n$ non-negative symmetric matrices with zero diagonal entries. Although $\Omega_{ijk}$ is non-convex, this sub-problem admits a closed-form solution:

**Case 1**: If $d_{ij} \leq \max\{d_{ik}, d_{jk}\}$, the constraint is already satisfied, and $D$ is feasible. Thus, $D_T = D \in \Omega_{ijk}$.

**Case 2**: If $d_{ij} > \max\{d_{ik}, d_{jk}\}$, suppose $d_{ik} \geq d_{jk}$. To satisfy the constraint $d_{ij}^T \leq d_{ik}^T$, $d_{ij}$ must be decreased to $d_{ij}^T$, and $d_{ik}$ must be increased to $d_{ik}^T$. Then the objective is

$$\|D_T - D\|_F^2 = 2 \cdot \left((d_{ij}^T - d_{ij})^2 + (d_{ik}^T - d_{ik})^2\right).$$

To minimize it, we set $d_{ij}^T = d_{ik}^T$. Let $\Delta = d_{ij} - d_{ij}^T \geq 0$, then the objective becomes:

$$2 \cdot (\Delta^2 + (d_{ij} - d_{ik} - \Delta)^2).$$

Minimizing this quadratic function gives $\Delta = \frac{1}{2}(d_{ij} - d_{ik})$.

As shown above, we obtain the matrix $D_T$ as follows:

$$d_{ij}^T = d_{ji}^T = d_{ij} - \Delta, \quad d_{ik}^T = d_{ki}^T = d_{ik} + \Delta,$$

while all other elements of $D_T$ remain unchanged from $D$.

The resulting matrix $D_T$ is denoted as $\mathcal{P}_{\Omega_{ijk}}(D)$, i.e., $\mathcal{P}_{\Omega_{ijk}}(D) := \operatorname{argmin}_{D_T \in \Omega_{ijk}} \|D_T - D\|_F^2$, representing the projection of $D$ onto the subset $\Omega_{ijk}$.

Next, we solve $\min_{D_T \in \Omega} \|D_T - D\|_F^2$, where $\Omega$ is defined as:

$$\Omega := \{D_T \in \mathbb{S}^n \mid d_{ij}^T \le \max\{d_{ik}^T, d_{jk}^T\}, \forall i, j, k \in [n]\}.$$

The feasible region $\Omega$ represents the ultrametric space and can be expressed as the intersection of all subsets: $\Omega = \cap_{ijk} \Omega_{ijk}$. The optimal solution is $\mathcal{P}_\Omega(D)$, the projection of $D$ onto $\Omega$, but finding this projection directly is challenging.

To find an approximate solution, we employ an iterative projection method called the Halpern-Lions-Wittmann-Bauschke (**HLWB**) projection (Censor, 2006; Li et al., 2023). Starting with $D^0 = D$, we iteratively refine $D^t$ by moving it toward $D$ and successively projecting it onto each subset $\Omega_{ijk}$. Empirically, each projection can enforce one strong triangle inequality, and we observe that iterative projections progressively satisfy more inequalities, potentially leading to an ultrametric. The iterative update rules are detailed below.

**Theorem 6** (**HLWB Projection** (Censor, 2006; Censor et al., 2022)). *Given subsets $\Omega_1, \cdots, \Omega_m$ in Euclidean space with a non-empty intersection $\Omega = \Omega_1 \cap \cdots \cap \Omega_m$, and a distance matrix $D \in \mathbb{R}^{n \times n}$, initialize $D^0 = D$. The iterative scheme for iteration $t = 1, 2, \cdots$ is as follows:*

*1. **Movement**: Compute $Y^t = \sigma_t D + (1 - \sigma_t) D^{t-1}$;*
*2. **Projection**: Update $D^t = \mathcal{P}_{\Omega_1} \mathcal{P}_{\Omega_2} \cdots \mathcal{P}_{\Omega_m}(Y^t)$,*

*where $\mathcal{P}_{\Omega_i}(Y)$ denotes the projection of $Y$ onto subset $\Omega_i$. If $\{\Omega_i\}_{i=1}^m$ are closed convex subsets and $\{\sigma_t\}_{t=1}^{+\infty}$ is a steering parameters sequence (i.e., $\sigma_t \in (0, 1)$ and $\sigma_t \to 0$), $D^t$ linearly converges to $\mathcal{P}_\Omega(D)$ as $t \to \infty$.*

The HLWB projection ensures convergence in convex problems, but such guarantees may not hold in this non-convex case. Nevertheless, a high-quality approximate solution $D^t$ can be achieved in just **one iteration** ($t = 1$). This method updates matrix entries directly and implicitly alters the tree structure after each projection, as a different $D_T$ induces a different tree $T$. The explicit tree topology is constructed only once in the final step of Algorithm 2, using the MST algorithm on the projected matrix $D^t$. The algorithm is presented below, with $\sigma_t = \frac{1}{t+1}$ following Li et al. (2023).

---

**Algorithm 2 UltraTWD-IP (Iterative Projection)**

**Input:** $D \in \mathbb{R}^{n \times n}$: cost matrix, $m$: maximum number of iterations (default $m = 1$).
**Output:** $T^*$: the ultrametric tree under the Frobenius norm.
1: Initialize $D^0 = D$.
2: **for** $t = 1$ to $m$ **do**
3:     Update $D^t \leftarrow \frac{1}{t+1} D + \frac{t}{t+1} D^{t-1}$   *(Movement step)*
4:     **for** each triplet $(i, j, k)$ **do**
5:         $D^t \leftarrow \mathcal{P}_{\Omega_{ijk}}(D^t)$         *(Projection step)*
6:     **end for**
7: **end for**
8: Construct the ultrametric tree: $T^* = \text{MST}(D^t)$.

---

## 3.5. Optimizing Ultrametrics with Frobenius Norm: Gradient Descent Method

The UltraTWD-IP updates $\frac{n(n-1)}{2}$ entries in $D_T \in \mathbb{R}^{n \times n}$ by projecting onto $O(n^3)$ subsets $\{\Omega_{ijk}\}$, which can be inefficient for large $n$. To improve efficiency and scalability, we adopt a **gradient descent (GD)** approach to directly optimize the tree structure, inspired by Chen et al. (2024).

When the tree structure is fixed, the ultrametric $D_T$ is fully determined by the node heights in $T$, as defined in Eq. (8):

$$D_T = [d_T(\boldsymbol{x_i}, \boldsymbol{x_j})] = [h(\text{LCA}(l_i, l_j))],$$

which defines a mapping $f : H_T \mapsto D_T$, where $H_T \in \mathbb{R}^{2n-1}$ denotes the heights of all $2n - 1$ nodes in $T$. By parameterizing $D_T$ with $H_T$, the objective becomes:

$$F(H_T) := \|f(H_T) - D\|_F^2,$$

reducing the number of parameters from $\frac{n(n-1)}{2}$ to $2n - 1$. The optimization process consists of the following steps. Initialize $T^0 = \text{MST}(D)$ with $H_T^0$. Then, for iteration $t$,

**1. Update $H_T$:** Fix the tree structure (i.e., the LCA relationships) and update $H_T^{t-1}$ via gradient descent:

$$H_T^t \leftarrow H_T^{t-1} - \alpha \nabla F(H_T^{t-1}),$$

where $\alpha$ is the learning rate. The gradient is efficiently computed using PyTorch's automatic differentiation.

**2. Update $D_T$:** Compute $D^t = f(H_T^t)$. Note that $D^t$ may have non-zero diagonal entries because $d_{ii}^t$ corresponds to the height of leaf node $l_i$, which can be non-zero in $H_T^t$. To ensure $D^t$ is a valid distance matrix, we adjust it as follows:

$$d_{ij}^t \leftarrow \frac{1}{2}(2d_{ij}^t - d_{ii}^t - d_{jj}^t),$$

which ensures zero diagonal values.

**3. Update $T$:** Apply the MST algorithm to $D^t$ to update the tree structure $T^t$ and obtain new heights $H_T^t$. Then repeat the update of $H_T^t$ in Step 1.

While based on UltraTree (Chen et al., 2024), our method uses an unsupervised objective $\|D_T - D\|_F^2$ instead of a supervised loss, resulting in a more efficient Algorithm 3.

---

**Algorithm 3 UltraTWD-GD (Gradient Descent)**

**Input:** $D \in \mathbb{R}^{n \times n}$: cost matrix, $m$: maximum iterations (default $m = 8$), $\alpha$: learning rate (default $\alpha = 0.02$).
**Output:** $T^*$: the ultrametric tree under the Frobenius norm
1: Initialize $T^0 = \text{MST}(D)$ with node heights $H_T^0$.
2: **for** $t = 1$ to $m$ **do**
3:     Update heights: $H_T^t \leftarrow H_T^{t-1} - \alpha \nabla F(H_T^{t-1})$.
4:     Compute the ultrametric: $D^t = f(H_T^t)$.
5:     Adjust entries of $D^t$: $d_{ij}^t \leftarrow \frac{1}{2}(2d_{ij}^t - d_{ii}^t - d_{jj}^t)$.
6:     Update tree: $T^t = \text{MST}(D^t)$ with heights $H_T^t$.
7: **end for**
8: Return the optimized ultrametric tree: $T^* = T^t$.

---

### 3.6. Algorithm Analysis

**Convergence Analysis.** Due to the non-convex and NP-hard nature of the problem (Eq. (11)), theoretical convergence is extremely hard to guarantee. Similar gradient descent methods in prior work (Chierchia & Perret, 2019; Chen et al., 2024) also lack convergence guarantees. Nevertheless, both UltraTWD-IP and GD methods exhibit **empirical convergence**, as demonstrated in Fig. 4.

**Time Complexity of Tree Learning.** Given a fixed support size $n$, both UltraTWD and UltraTree learn a single tree. **(1) UltraTWD-MST:** Total complexity is $\mathcal{O}(n^2)$, as computing the MST for $D \in \mathbb{R}^{n \times n}$ takes $\mathcal{O}(n^2)$ time. **(2) UltraTWD-IP:** It involves $\mathcal{O}(n^3)$ projections with $\mathcal{O}(1)$ cost each. We use **only one iteration** in practice, resulting in $\mathcal{O}(n^3)$ total complexity, which is practical for moderately large $n$. **(3) UltraTWD-GD:** Each iteration takes $\mathcal{O}(n^2)$ due to gradient descent and MST, giving $\mathcal{O}(mn^2)$ overall. We set $m = 8$ in practice, making it more scalable than UltraTWD-IP for large $n$. **(4) UltraTree (Chen et al., 2024):** Precomputing $W_1$ for $M$ training pairs requires $\mathcal{O}(Mn^3 \log n)$. Tree learning has $\mathcal{O}(mn^2 + mMn)$ complexity due to computing $W_T$ in each iteration for loss evaluation. Overall, UltraTWD-GD is significantly more efficient than UltraTree.

*Table 1.* Time Complexity Comparison. For $N$ distributions over a support of size $n$, "W Computation" is the cost of computing the pairwise distance matrix $W_T$ or $W_1 \in \mathbb{R}^{N \times N}$.

| Time Complexity | Tree Learning | W Computation |
|---|---|---|
| UltraTWD-MST | $\mathcal{O}(n^2)$ | $\mathcal{O}(N^2 \cdot n)$ |
| UltraTWD-IP | $\mathcal{O}(n^3)$ | $\mathcal{O}(N^2 \cdot n)$ |
| UltraTWD-GD | $\mathcal{O}(mn^2)$ | $\mathcal{O}(N^2 \cdot n)$ |
| UltraTree | $\mathcal{O}(mn^2 + mMn)$ | $\mathcal{O}(N^2 \cdot n)$ |
| $W_1$ | – | $\mathcal{O}(N^2 \cdot n^3 \log n)$ |

**Time Complexity of $W$ Computation.** For $N$ distributions over a support of size $n$, computing the 1-Wasserstein distance matrix $W_1 \in \mathbb{R}^{N \times N}$ requires $\mathcal{O}(N^2 \cdot n^3 \log n)$ time, while computing the $W_T$ matrix on an ultrametric tree requires only $\mathcal{O}(N^2 \cdot n)$ time. For a fair comparison, including UltraTWD's tree learning cost ($\mathcal{O}(n^3)$ or $\mathcal{O}(mn^2)$), our methods remain significantly faster than exact $W_1$ computation, especially for large datasets. Results in Section 4.4 further demonstrate the computational efficiency of TWD.

## 4. Experiments

### 4.1. Experimental Setting

**Dataset.** We evaluate our methods using four benchmark text datasets: BBCSport, Reuters, Ohsumed, and Recipe (Huang et al., 2016), following previous studies (Takezawa et al., 2021; Chen et al., 2024). These datasets are publicly available[1], with detailed statistics provided in Table 2.

[1]https://github.com/mkusner/wmd

*Table 2.* Statistical Information of Text Datasets.

| Dataset | Avg. Words ($n$) | # Test Data ($N$) | # Classes |
|---|---|---|---|
| BBCSport | 6,051 | 220 | 5 |
| Reuters | 6,416 | 1,000 | 8 |
| Ohsumed | 9,467 | 1,000 | 10 |
| Recipe | 4,084 | 1,311 | 15 |

**Implementation.** Each dataset contains 5 test sets. For each test set, we construct the vocabulary $X = [\boldsymbol{x_1}, \ldots, \boldsymbol{x_n}] \in \mathbb{R}^{d \times n}$, where $d = 300$ is the word embedding dimension and $n$ is the number of unique words (average values shown in Table 2). Each test document $\mu_i$ is represented as a normalized bag-of-words distribution. We compute the cost matrix $D \in \mathbb{R}^{n \times n}$ using Euclidean distances: $d_{ij} = \|\boldsymbol{x_i} - \boldsymbol{x_j}\|_2$. The vocabulary $X$ and cost $D$ are then used to learn the tree $T$, and each $W_T(\mu_i, \mu_j)$ is computed based on $T$.

**Evaluation.** The effectiveness of the tree-Wasserstein distance is assessed based on the following objectives:
*(1) Approximation Accuracy:* Measuring the error between tree-Wasserstein distance and 1-Wasserstein distance.
*(2) Application Performance:* Evaluating the utility of tree-Wasserstein distance in text-based tasks, including document retrieval, ranking, and classification.
*(3) Performance Analysis:* Analyzing the impact of hyperparameters and the algorithm efficiency trade-off.

**Baseline Methods.** We compare the UltraTWD methods with 10 approximation methods across various categories. Detailed descriptions are provided in **Appendix A.4**.
*(1) Entropy-based method:* **Sinkhorn** distance (Cuturi, 2013) is implemented using the POT library (Flamary et al., 2021), with a regularization parameter $\lambda = 1$ and a maximum of 100 iterations.[2]
*(2) Tree-construction methods:* **QuadTree** (Indyk & Thaper, 2003) and **ClusterTree** (Le et al., 2019), implemented by modifying the code from Yamada et al. (2022).
*(3) Weight-optimized methods:* Weight-optimized QuadTree (**qTWD**) and ClusterTree (**cTWD**) with a regularization parameter $\lambda = 0.001$ (Yamada et al., 2022).[3]
*(4) Tree-sliced methods:* **Sliced-QuadTree**, **Sliced-ClusterTree**, **Sliced-qTWD**, and **Sliced-cTWD**, average TWD over 3 randomly sampled trees (Yamada et al., 2022).
*(5) Supervised method:* **UltraTree** (Chen et al., 2024) is trained on 1,000 randomly generated distributions with 1% sparsity and precomputed 1-Wasserstein distances.[4]

All experiments were conducted using Python 3.8 on a Linux server with an AMD EPYC 7742 64-Core Processor, 256 logical CPUs, and 512 GB RAM. The average performance and standard deviation are reported across the 5 test sets. Additional details can be found in **Appendix B**.

[2]Following Chen et al. (2024) to balance runtime and accuracy.
[3]https://github.com/oist/treeOT
[4]https://github.com/chens5/tree_learning

*Table 3.* Comprehensive comparison of tree-Wasserstein distance methods across four text datasets. **Bold** indicates the best result, underline marks the second-best, and wavy underline denotes the third-best. Our UltraTWD-IP and GD methods outperform the baseline methods in most cases across (a) approximation error, (b) document retrieval, (c) document ranking, and (d) document classification.

| Metric | (a) RE-W ↓: Approximation Error of $W_T$ | | | | (b) Precision ↑: Document Retrieval | | | |
| Dataset | BBCSport | Reuters | Ohsumed | Recipe | BBCSport | Reuters | Ohsumed | Recipe |
| --- | --- | --- | --- | --- | --- | --- | --- | --- |
| Sinkhorn | $0.258_{\pm0.003}$ | $0.167_{\pm0.003}$ | $0.166_{\pm0.002}$ | $0.244_{\pm0.001}$ | $0.534_{\pm0.009}$ | $0.412_{\pm0.011}$ | $0.354_{\pm0.012}$ | $0.207_{\pm0.011}$ |
| QuadTree | $0.643_{\pm0.080}$ | $0.630_{\pm0.036}$ | $0.649_{\pm0.089}$ | $0.745_{\pm0.039}$ | $0.722_{\pm0.026}$ | $0.732_{\pm0.015}$ | $0.552_{\pm0.042}$ | $0.694_{\pm0.028}$ |
| ClusterTree | $0.774_{\pm0.003}$ | $0.733_{\pm0.012}$ | $0.768_{\pm0.007}$ | $0.736_{\pm0.009}$ | $0.548_{\pm0.017}$ | $0.543_{\pm0.011}$ | $0.315_{\pm0.014}$ | $0.395_{\pm0.016}$ |
| qTWD | $0.162_{\pm0.004}$ | $0.102_{\pm0.001}$ | $0.126_{\pm0.003}$ | $0.139_{\pm0.001}$ | $0.810_{\pm0.004}$ | $0.819_{\pm0.007}$ | $0.691_{\pm0.010}$ | $0.795_{\pm0.010}$ |
| cTWD | $0.116_{\pm0.002}$ | $0.079_{\pm0.002}$ | $0.097_{\pm0.004}$ | $0.106_{\pm0.004}$ | $0.852_{\pm0.004}$ | $0.840_{\pm0.002}$ | $0.723_{\pm0.004}$ | $0.820_{\pm0.002}$ |
| Sliced-QuadTree | $0.689_{\pm0.042}$ | $0.651_{\pm0.016}$ | $0.681_{\pm0.034}$ | $0.758_{\pm0.021}$ | $0.795_{\pm0.007}$ | $0.801_{\pm0.004}$ | $0.655_{\pm0.009}$ | $0.769_{\pm0.005}$ |
| Sliced-ClusterTree | $0.771_{\pm0.001}$ | $0.734_{\pm0.009}$ | $0.770_{\pm0.004}$ | $0.747_{\pm0.007}$ | $0.637_{\pm0.015}$ | $0.652_{\pm0.008}$ | $0.445_{\pm0.018}$ | $0.522_{\pm0.013}$ |
| Sliced-qTWD | $0.164_{\pm0.002}$ | $0.103_{\pm0.001}$ | $0.127_{\pm0.002}$ | $0.140_{\pm0.001}$ | $0.821_{\pm0.005}$ | $0.824_{\pm0.002}$ | $0.708_{\pm0.006}$ | $0.807_{\pm0.004}$ |
| Sliced-cTWD | $0.116_{\pm0.002}$ | $0.079_{\pm0.001}$ | $0.095_{\pm0.001}$ | $0.102_{\pm0.003}$ | $\underset{\sim}{0.863}_{\pm0.002}$ | $0.849_{\pm0.002}$ | $0.742_{\pm0.005}$ | $0.831_{\pm0.002}$ |
| UltraTree | $\underline{0.022}_{\pm0.001}$ | $\underline{0.041}_{\pm0.003}$ | $\underline{0.038}_{\pm0.001}$ | $\underline{0.027}_{\pm0.005}$ | $0.842_{\pm0.006}$ | $0.834_{\pm0.002}$ | $0.749_{\pm0.003}$ | $0.830_{\pm0.004}$ |
| UltraTWD-MST | $0.109_{\pm0.001}$ | $0.050_{\pm0.004}$ | $0.077_{\pm0.004}$ | $0.098_{\pm0.007}$ | $0.860_{\pm0.005}$ | $\underline{0.854}_{\pm0.003}$ | $\underset{\sim}{0.765}_{\pm0.003}$ | $\underline{0.840}_{\pm0.003}$ |
| UltraTWD-IP | $\mathbf{0.014}_{\pm0.000}$ | $\underline{0.033}_{\pm0.001}$ | $\underline{0.036}_{\pm0.001}$ | $\mathbf{0.026}_{\pm0.000}$ | $\mathbf{0.885}_{\pm0.002}$ | $\mathbf{0.876}_{\pm0.003}$ | $\mathbf{0.788}_{\pm0.005}$ | $\mathbf{0.866}_{\pm0.001}$ |
| UltraTWD-GD | $\underset{\sim}{0.028}_{\pm0.001}$ | $\mathbf{0.030}_{\pm0.001}$ | $\mathbf{0.017}_{\pm0.000}$ | $\underset{\sim}{0.035}_{\pm0.001}$ | $\underline{0.867}_{\pm0.005}$ | $0.860_{\pm0.002}$ | $\underline{0.776}_{\pm0.003}$ | $\underset{\sim}{0.848}_{\pm0.002}$ |

| Metric | (c) MRR ↑: Document Ranking | | | | (d) Accuracy ↑: Document Classification | | | |
| Dataset | BBCSport | Reuters | Ohsumed | Recipe | BBCSport | Reuters | Ohsumed | Recipe |
| --- | --- | --- | --- | --- | --- | --- | --- | --- |
| Sinkhorn | $0.533_{\pm0.017}$ | $0.389_{\pm0.005}$ | $0.395_{\pm0.010}$ | $0.226_{\pm0.011}$ | $0.359_{\pm0.000}$ | $0.773_{\pm0.008}$ | $0.292_{\pm0.011}$ | $0.381_{\pm0.005}$ |
| QuadTree | $0.789_{\pm0.021}$ | $0.790_{\pm0.016}$ | $0.662_{\pm0.036}$ | $0.761_{\pm0.026}$ | $0.805_{\pm0.010}$ | $0.889_{\pm0.014}$ | $0.398_{\pm0.021}$ | $0.488_{\pm0.008}$ |
| ClusterTree | $0.588_{\pm0.025}$ | $0.603_{\pm0.014}$ | $0.377_{\pm0.021}$ | $0.452_{\pm0.024}$ | $0.771_{\pm0.041}$ | $0.891_{\pm0.014}$ | $0.406_{\pm0.027}$ | $0.461_{\pm0.007}$ |
| qTWD | $0.865_{\pm0.009}$ | $0.865_{\pm0.006}$ | $0.783_{\pm0.014}$ | $0.852_{\pm0.011}$ | $0.819_{\pm0.020}$ | $0.892_{\pm0.011}$ | $0.394_{\pm0.027}$ | $0.493_{\pm0.009}$ |
| cTWD | $0.884_{\pm0.015}$ | $0.882_{\pm0.007}$ | $0.810_{\pm0.009}$ | $0.873_{\pm0.011}$ | $0.809_{\pm0.022}$ | $0.896_{\pm0.014}$ | $0.411_{\pm0.026}$ | $0.493_{\pm0.007}$ |
| Sliced-QuadTree | $0.846_{\pm0.013}$ | $0.848_{\pm0.009}$ | $0.754_{\pm0.013}$ | $0.827_{\pm0.007}$ | $0.823_{\pm0.016}$ | $0.892_{\pm0.015}$ | $0.404_{\pm0.025}$ | $0.492_{\pm0.007}$ |
| Sliced-ClusterTree | $0.676_{\pm0.023}$ | $0.715_{\pm0.008}$ | $0.515_{\pm0.015}$ | $0.585_{\pm0.018}$ | $0.798_{\pm0.027}$ | $\underline{0.902}_{\pm0.009}$ | $\underset{\sim}{0.423}_{\pm0.027}$ | $0.476_{\pm0.008}$ |
| Sliced-qTWD | $0.872_{\pm0.011}$ | $0.869_{\pm0.006}$ | $0.796_{\pm0.010}$ | $0.862_{\pm0.011}$ | $0.819_{\pm0.019}$ | $0.893_{\pm0.013}$ | $0.396_{\pm0.028}$ | $0.494_{\pm0.007}$ |
| Sliced-cTWD | $0.890_{\pm0.015}$ | $0.887_{\pm0.005}$ | $0.819_{\pm0.010}$ | $0.880_{\pm0.009}$ | $0.817_{\pm0.022}$ | $0.897_{\pm0.013}$ | $0.412_{\pm0.026}$ | $0.494_{\pm0.006}$ |
| UltraTree | $0.890_{\pm0.009}$ | $0.877_{\pm0.008}$ | $0.831_{\pm0.010}$ | $0.877_{\pm0.012}$ | $0.826_{\pm0.017}$ | $0.894_{\pm0.014}$ | $0.416_{\pm0.023}$ | $\mathbf{0.495}_{\pm0.007}$ |
| UltraTWD-MST | $\underset{\sim}{0.909}_{\pm0.008}$ | $\underline{0.893}_{\pm0.005}$ | $\underline{0.845}_{\pm0.008}$ | $\underset{\sim}{0.886}_{\pm0.010}$ | $\underset{\sim}{0.835}_{\pm0.016}$ | $\underset{\sim}{0.899}_{\pm0.013}$ | $0.421_{\pm0.024}$ | $\mathbf{0.495}_{\pm0.007}$ |
| UltraTWD-IP | $\mathbf{0.924}_{\pm0.011}$ | $\mathbf{0.913}_{\pm0.005}$ | $\mathbf{0.863}_{\pm0.005}$ | $\mathbf{0.908}_{\pm0.004}$ | $\mathbf{0.839}_{\pm0.017}$ | $\mathbf{0.905}_{\pm0.010}$ | $\mathbf{0.428}_{\pm0.020}$ | $\mathbf{0.495}_{\pm0.006}$ |
| UltraTWD-GD | $\underline{0.922}_{\pm0.009}$ | $\underset{\sim}{0.899}_{\pm0.006}$ | $\underset{\sim}{0.855}_{\pm0.010}$ | $\underline{0.895}_{\pm0.007}$ | $\underline{0.836}_{\pm0.015}$ | $\underset{\sim}{0.899}_{\pm0.013}$ | $\underline{0.427}_{\pm0.024}$ | $\mathbf{0.495}_{\pm0.007}$ |

## 4.2. Approximation Error of Tree-Wasserstein Distance

Each test set contains $N$ documents $\{\mu_1, \ldots, \mu_N\}$ represented as normalized bag-of-words distributions. We compute the pairwise 1-Wasserstein distance matrix $W_1 = [W_1(\mu_i, \mu_j)] \in \mathbb{R}^{N \times N}$ as the ground truth and the tree-Wasserstein distance matrix $W_T = [W_T(\mu_i, \mu_j)] \in \mathbb{R}^{N \times N}$ as its approximation (including the Sinkhorn method). We evaluate the **Relative Error** of $W_T$ (**RE-W**), defined as:

$$\text{RE-W} = \frac{\|W_T - W_1\|_F}{\|W_1\|_F}.$$

Table 3(a) shows that our UltraTWD-IP approach achieves the smallest RE-W compared to all baseline methods.

• **Comparison with Unsupervised Methods:**
**(1) Entropy-based Method:** Sinkhorn distance exhibits higher errors due to entropy regularization, which improves speed at the cost of accuracy. A smaller regularization parameter $\lambda$ and more iterations can improve accuracy but significantly slow down the computation. For example, on the Recipe dataset, setting $\lambda = 0.01$ with 300 iterations reduces RE-W from 0.244 to 0.002, but runtime rises from 0.1 to 2.2 hours, while UltraTWD-IP completes in 1.2 hours.

**(2) Tree-construction Method:** QuadTree and ClusterTree exhibit large approximation errors (e.g., 0.643 and 0.774 on the BBCSport dataset) because their edge weights are fixed based solely on tree depth, resulting in suboptimal tree representations and poor accuracy.
**(3) Weight-optimized Method:** The qTWD and cTWD methods enhance QuadTree and ClusterTree by optimizing edge weights, significantly reducing the relative error on the BBCSport dataset from 0.643 (QuadTree) and 0.774 (ClusterTree) to 0.162 (qTWD) and 0.116 (cTWD), respectively.
**(4) Tree-sliced Method:** Sliced methods average TWDs over multiple random trees, yielding comparable or slightly higher errors due to the variance from multiple trees.

Overall, unsupervised baselines cannot optimize tree structures and instead rely on randomly constructed trees from QuadTree or ClusterTree, resulting in suboptimal accuracy. In contrast, our methods optimize both the tree structure and edge weights by refining the ultrametric $D_T$. Up to isomorphism, each $D_T$ corresponds to a unique tree structure with specific edge weights. By aligning $D_T$ closely with the cost matrix $D$, all three UltraTWD methods dynamically adjust the tree structure, achieving lower errors across all four datasets compared to unsupervised baseline methods.

*Table 4.* Approximation error of UltraTree and UltraTWD methods. RE-W $= \frac{\|W_T - W_1\|_F}{\|W_1\|_F}$ and RE-D $= \frac{\|D_T - D\|_F}{\|D\|_F}$. **Bold** is the best result and underline is the second-best.

| Dataset | Reuters | | Ohsumed | |
|---|---|---|---|---|
| Metric | RE-W | RE-D | RE-W | RE-D |
| UltraTree-1% | $0.041_{\pm0.003}$ | $0.113_{\pm0.001}$ | $0.038_{\pm0.001}$ | $0.123_{\pm0.001}$ |
| UltraTree-2% | $0.061_{\pm0.002}$ | $0.130_{\pm0.000}$ | $0.069_{\pm0.001}$ | $0.147_{\pm0.000}$ |
| UltraTree-10% | $0.127_{\pm0.003}$ | $0.187_{\pm0.001}$ | $0.119_{\pm0.001}$ | $0.189_{\pm0.000}$ |
| UltraTWD-IP | $0.033_{\pm0.001}$ | $0.102_{\pm0.001}$ | $0.036_{\pm0.001}$ | $0.115_{\pm0.001}$ |
| UltraTWD-GD | **$0.030_{\pm0.001}$** | **$0.100_{\pm0.001}$** | **$0.017_{\pm0.000}$** | **$0.101_{\pm0.000}$** |

• **Comparison with Supervised Methods:** UltraTree, a state-of-the-art baseline, learns the ultrametric $D_T$ by minimizing the loss $\|W_T^{\text{train}} - W_1^{\text{train}}\|_F^2$ on training data. However, its performance is highly sensitive to the sparsity of training distributions, which limits its robustness. As shown in Table 4, the approximation error is low at 1% sparsity, as it matches the average sparsity in the test data. When sparsity increases to 2% or 10%, the RE-W on the Reuters dataset rises from 0.041 to 0.061 and 0.127, respectively, along with higher RE-D. This indicates that UltraTree struggles to generalize when the sparsity levels between training and test distributions differ.

In comparison, our UltraTWD-IP and GD methods are inherently robust to sparsity, as they do not require training data. Instead of minimizing $\|W_T^{\text{train}} - W_1^{\text{train}}\|_F^2$, we minimize the fundamental difference $\|D_T - D\|_F^2$, resulting in lower RE-D and RE-W compared to UltraTree.

### 4.3. Applications of Tree-Wasserstein Distance

We further evaluate the quality of tree-Wasserstein distance matrices in document retrieval, ranking, and classification, showing the utility of our methods in text applications.

**Document Retrieval.** We evaluate performance on a nearest neighbor retrieval task. For each test document, the top-10 nearest neighbors among the remaining $N - 1$ documents are retrieved using both $W_T$ and $W_1$, resulting in two sets: $I_T$ and $I_1$, respectively. We use **Precision** as the evaluation metric, defined as Precision $= \frac{|I_T \cap I_1|}{10}$. Table 3(b) shows that our methods achieve the highest precision, following the order: UltraTWD-IP > UltraTWD-GD > UltraTWD-MST > most baselines. This demonstrates that UltraTWD effectively preserves neighborhood structures, ensuring that similar documents under $W_1$ remain close under $W_T$. Visualizations are provided in **Appendix C.1**.

**Document Ranking.** We assess ranking performance using the Mean Reciprocal Rank (**MRR**), calculated as MRR $= \frac{1}{N}\sum_{i=1}^{N}\frac{1}{\text{rank}_i}$. For a given query document $\mu_i$, $\text{rank}_i$ is the position of the first relevant item in $W_T$ based on the ranking induced by $W_1$. MRR reflects the average ranking quality across $N$ queries, ranging from 0 (worst) to 1 (best). Table 3(c) shows that our methods achieve the highest MRR, with relevant items consistently ranked near the top.

**Document Classification.** For text classification, we compute the kernel $K = \exp(-W_T/\sigma)$, where $\sigma = \text{median}(W_T)$, and apply a kernel SVM with 10-fold cross-validation. As shown in Table 3(d), UltraTWD-IP achieves the highest classification accuracy across all four datasets, showing its practical advantage for downstream tasks.

### 4.4. Performance Analysis

**Hyperparameter and Convergence Analysis.** Fig. 4 shows the impact of iterations and learning rates on RE-D convergence. Both UltraTWD-IP and GD methods demonstrate empirical convergence, though not necessarily to a global minimum. **(1) UltraTWD-IP:** It converges in fewer iterations and achieves higher retrieval precision. In practice, one iteration is often sufficient. **(2) UltraTWD-GD:** A learning rate of 0.05 accelerates RE-D reduction but may overshoot local minima, leading to unstable precision. A learning rate of 0.02 provides stable convergence and high precision, and is used in all experiments. Additional results are provided in **Appendix C.2**.

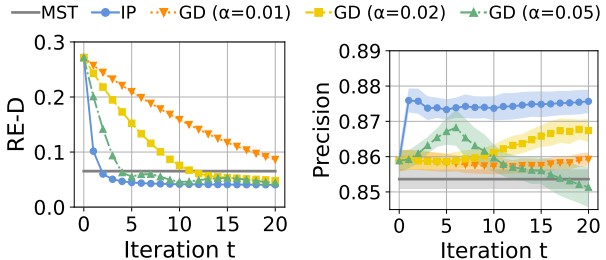

*Figure 4.* Hyperparameter analysis of UltraTWD methods on the Reuters dataset, illustrating empirical convergence.

**Efficiency Analysis of Learning Time.** Table 5 reports the average time for tree learning. UltraTWD-MST is the fastest (as fast as ClusterTree), completing in just 2–7 seconds, while achieving much lower errors than all unsupervised baselines. UltraTWD-GD offers a great balance between efficiency and accuracy, with learning times of 15-72 seconds, comparable to Sliced-qTWD/cTWD, but achieving much smaller errors and higher retrieval precision. Compared to UltraTree, UltraTWD-GD is up to 43× faster and also more accurate, making it well-suited for real-world applications.

**Efficiency Analysis of $W$ Computation.** We generate 100 random distributions over the BBCSport vocabulary (average 6,051 words), each containing $n_{\text{valid}}$ words (non-zero entries). We compare the total time required to compute the matrix $W_1$ or $W_T \in \mathbb{R}^{100 \times 100}$, including tree learning time. As shown in Table 6, when $n_{\text{valid}} = 5,000$, computing the $W_1$ matrix takes 5.0 hours, with each $W_1(\mu, \nu)$ requiring 3.6 seconds. In contrast, UltraTWD variants complete the $W_T$ matrix within 1 hour, with each $W_T(\mu, \nu)$ computed in just 0.5 seconds, achieving a significant speedup. UltraTree is slower than UltraTWD due to its costly training process, including the precomputation of $W_1$ for training pairs.

_Table 5._ Tree learning time (in seconds) of TWD methods.

| Dataset | BBCSport | Reuters | Ohsumed | Recipe |
|---|---|---|---|---|
| Average Words ($n$) | 6,051 | 6,416 | 9,467 | 4,084 |
| QuadTree | $8_{\pm 0}$ | $9_{\pm 0}$ | $13_{\pm 0}$ | $6_{\pm 0}$ |
| ClusterTree | $3_{\pm 0}$ | $3_{\pm 0}$ | $5_{\pm 0}$ | $2_{\pm 0}$ |
| qTWD | $14_{\pm 1}$ | $14_{\pm 2}$ | $20_{\pm 2}$ | $10_{\pm 2}$ |
| cTWD | $13_{\pm 1}$ | $13_{\pm 2}$ | $15_{\pm 1}$ | $11_{\pm 1}$ |
| Sliced-QuadTree | $27_{\pm 1}$ | $27_{\pm 1}$ | $41_{\pm 2}$ | $16_{\pm 0}$ |
| Sliced-ClusterTree | $10_{\pm 1}$ | $10_{\pm 0}$ | $16_{\pm 1}$ | $6_{\pm 0}$ |
| Sliced-qTWD | $40_{\pm 2}$ | $39_{\pm 3}$ | $55_{\pm 3}$ | $28_{\pm 3}$ |
| Sliced-cTWD | $37_{\pm 1}$ | $37_{\pm 3}$ | $42_{\pm 2}$ | $31_{\pm 3}$ |
| UltraTree | $1217_{\pm 464}$ | $938_{\pm 39}$ | $1494_{\pm 171}$ | $361_{\pm 83}$ |
| UltraTWD-MST | $3_{\pm 0}$ | $3_{\pm 0}$ | $7_{\pm 0}$ | $2_{\pm 0}$ |
| UltraTWD-IP | $1207_{\pm 77}$ | $1437_{\pm 65}$ | $4583_{\pm 88}$ | $373_{\pm 4}$ |
| UltraTWD-GD | $28_{\pm 7}$ | $34_{\pm 11}$ | $72_{\pm 21}$ | $15_{\pm 3}$ |

_Table 6._ Total time (in hours) of computing $W_1$ or $W_T \in \mathbb{R}^{100 \times 100}$ on the BBCSport dataset, including tree learning time. $n_{\text{valid}}$ represents the number of valid words in each distribution.

| Valid Words ($n_{\text{valid}}$) | 1,000 | 2,000 | 3,000 | 4,000 | 5,000 |
|---|---|---|---|---|---|
| $W_1$ | 0.2 | 0.9 | 2.3 | 3.6 | 5.0 |
| UltraTree | 0.7 | 0.8 | 1.0 | 1.1 | 1.1 |
| UltraTWD-MST | 0.2 | 0.4 | 0.5 | 0.6 | 0.7 |
| UltraTWD-IP | 0.6 | 0.7 | 0.9 | 1.0 | 1.0 |
| UltraTWD-GD | 0.2 | 0.4 | 0.5 | 0.6 | 0.7 |

### 4.5. Qualitative Comparison of Tree Structures

To qualitatively assess the semantic interpretability of the learned trees, we select 7 representative words from the Recipe dataset, grouped into 4 semantic categories: Cooking (_Bake_, _Fry_), Fruit (_Apple_, _Watermelon_), Meat (_Chicken_, _Beef_), and Vegetable (_Lettuce_). The word embedding matrix $X \in \mathbb{R}^{7 \times 300}$ and the Euclidean distance matrix $D \in \mathbb{R}^{7 \times 7}$ are used as input. Fig. 5 presents the resulting tree structures. **(a) QuadTree & qTWD** produce flat and shallow trees without a semantic structure. **(b) ClusterTree & cTWD** partially group related words but often mix categories. In contrast, **(c) UltraTWD** clearly separates all four semantic groups and produces a well-structured hierarchy. In this simple case, all three UltraTWD variants yield the same tree, demonstrating UltraTWD's strength in capturing semantically meaningful structures that reflect underlying word relationships.

### 5. Conclusion

We introduce UltraTWD, a novel unsupervised framework for tree-Wasserstein distances that jointly optimizes the tree structure and edge weights, addressing the limitations of existing methods. By solving ultrametric nearness problems, we develop new algorithms leveraging minimum spanning trees, iterative projection, and gradient descent. These methods demonstrate superior accuracy in 1-Wasserstein distance estimation and significantly improve performance in tasks such as document retrieval, ranking, and classification, making them valuable for Wasserstein-based applications.

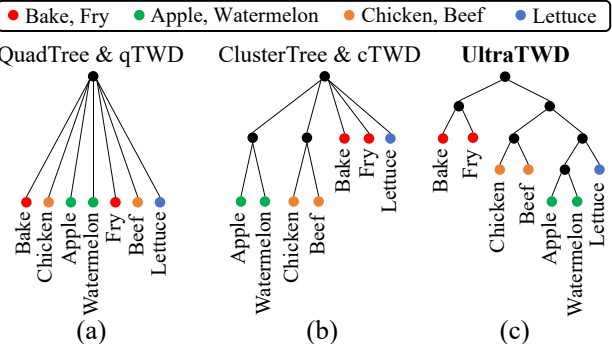

_Figure 5._ Tree structures produced by UltraTWD and baseline methods for 7 semantically grouped words from the Recipe dataset.

## Acknowledgments

The work of Fangchen Yu, Jianfeng Mao and Wenye Li was supported in part by Shenzhen Science and Technology Program (No. ZDSYS20230626091302006), in part by the National Natural Science Foundation of China (No. 72394362 and U1733102), in part by the Guangdong Provincial Key Laboratory of Big Data Computing, The Chinese University of Hong Kong, Shenzhen (No. 2024SC0003), in part by the Shenzhen Science and Technology Innovation Committee under the Shenzhen Stability Science Program grant (No. 2024SC0010), in part by the Shenzhen Research Institute of Big Data (No. J00120250001), and in part by CUHK-Shenzhen (No. PF.01.000404). The work of Fangchen Yu and Qiang Sun was supported in part by the Natural Sciences and Engineering Research Council of Canada (Grant RGPIN-2018-06484), computing resources provided by the Digital Research Alliance of Canada, and MBZUAI.

## Impact Statement

This paper presents UltraTWD, a novel framework for tree-Wasserstein distance that aims to advance the field of machine learning by improving the efficiency and accuracy of 1-Wasserstein distance approximations. The proposed methods have the potential to benefit a wide range of applications, including text retrieval, ranking, and classification tasks, by offering accurate and reliable solutions.

Overall, we believe UltraTWD contributes positively to the field of machine learning and offers practical benefits across various domains, with no immediate ethical concerns requiring specific attention.

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

# A. Background and Related Work

## A.1. Background of Wasserstein Distance

The Wasserstein distance (Villani, 2008) measures the distance between probability distributions. Consider two discrete measures $\mu = \sum_{i=1}^{n} a_i \delta_{x_i}$ and $\nu = \sum_{j=1}^{n} b_j \delta_{x_j}$, represented as 1-dimensional histograms shown below:

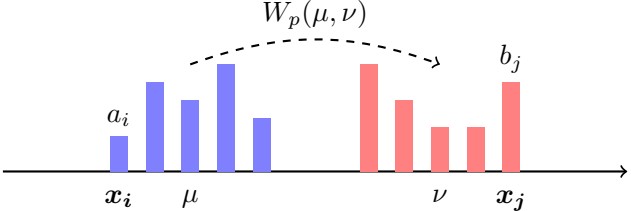

The $p$-Wasserstein distance is defined as

$$W_p(\mu, \nu) := \left( \min_{\gamma \in \Gamma(\mu,\nu)} \sum_{i,j=1}^{n} \gamma_{ij} d(x_i, x_j)^p \right)^{1/p},$$

where $\gamma$ is the transport plan in the set $\Gamma(\mu, \nu)$ defined as

$$\Gamma(\mu, \nu) = \{ \gamma \in \mathbb{R}_+^{n \times n} \mid \sum_j \gamma_{ij} = a_i, \sum_i \gamma_{ij} = b_j, \forall i, j \}.$$

In this paper, we focus on the commonly used 1-Wasserstein distance, defined as

$$W_1(\mu, \nu) := \min_{\gamma \in \Gamma(\mu,\nu)} \sum_{i,j=1}^{n} \gamma_{ij} d(x_i, x_j),$$

which is also known as the optimal transport distance or Earth Mover's Distance (Pele & Werman, 2009).

At a high level, the cost $d(x_i, x_j)$ represents the cost of transporting a unit mass from $x_i$ to $x_j$. Thus, the 1-Wasserstein distance computes the minimum cost required to transport all mass from $\{a_i\}$ to $\{b_j\}$, where $\gamma_{ij}$ specifies the amount of mass moved from $x_i$ to $x_j$.

In text analysis, $x_i$ denotes the embedding vector of the $i$-th word in the vocabulary, and the vocabulary of $n$ words is represented by a word embedding matrix $X = [x_1, x_2, \ldots, x_n] \in \mathbb{R}^{d \times n}$. A document $\mu$ is modeled as a normalized bag-of-words distribution over the vocabulary $X$, expressed as $\mu = \sum_{i=1}^{n} a_i \delta_{x_i}$, where $a_i$ is the frequency of the $i$-th word in the document. The normalization ($\sum_{i=1}^{n} a_i = 1$) ensures $\mu$ is a valid probability distribution.

When the Euclidean distance between word vectors is used as the cost, i.e., $d(x_i, x_j) = \|x_i - x_j\|_2$, the 1-Wasserstein distance corresponds to the Word Mover's Distance (WMD) (Kusner et al., 2015). Intuitively, WMD measures the minimum cost needed to transport the words from one document to another.

Despite its effectiveness, the 1-Wasserstein distance has a computational complexity of $\mathcal{O}(n^3 \log n)$, making it computationally expensive in practice (Villani, 2008).

## A.2. Background of Tree-Wasserstein Distance

**Motivation of Tree-Wasserstein Distance.** The 1-Wasserstein distance requires solving a linear programming problem to obtain the optimal transport plan $\gamma$, which can be computationally expensive. To improve the efficiency, the tree-Wasserstein distance encodes the cost $d(x_i, x_j)$ into the shortest path distance $d_T(x_i, x_j)$ defined on the tree $T$. Then the tree-Wasserstein distance defined on the tree has a closed-form solution:

$$W_T(\mu, \nu) = \sum_{e \in T} w_e \cdot |\mu(T_e) - \nu(T_e)|, \tag{13}$$

where $w_e$ is the weight of each edge $e$ in the tree. The derivation of this formula can be found in Proposition 1 of Le et al. (2019).

**Linear Time Complexity.** The closed-form solution given in Eq. (13) requires summing over all edges in the tree. In a rooted tree, the number of edges is equal to the number of nodes, denoted by $n_{\text{node}}$. Therefore, the computation of the tree-Wasserstein distance has a time complexity of $\mathcal{O}(n_{\text{node}})$. Since the number of nodes is bounded by $\mathcal{O}(n)$, where $n$ represents the number of leaf nodes, the overall complexity is also $\mathcal{O}(n)$ (Chen et al., 2024). Consequently, the tree-Wasserstein distance can be computed in linear time with respect to the number of leaf nodes.

## A.3. Background of Ultrametric

**Definition of Ultrametric.** An ultrametric $D \in \mathbb{R}^{n \times n}$ is a special type of tree metric that satisfies the strong triangle inequality: $d_{ij} \leq \max\{d_{ik}, d_{jk}\}$ for all $i, j, k$. This implies that any triplet $(i, j, k)$ forms an isosceles triangle. Ultrametrics naturally correspond to rooted trees with $n$ leaf nodes, where all leaves are equidistant from the root. In such trees, the distance between two leaves is defined by the height of their lowest common ancestor. The resulting pairwise distance matrix among the leaves is an ultrametric matrix $D$. Note that ultrametric trees are rooted but not necessarily binary. In practice, we follow Chen et al. (2024) and use the hierarchical minimum spanning tree procedure (Prim, 1957) to construct binary ultrametric trees.

**Benefits of Ultrametrics.** Ultrametrics are preferred over general tree metrics in our setting for several reasons. First, constructing a tree from a general tree metric is nontrivial, while ultrametric trees can be efficiently generated using algorithms such as the minimum spanning tree. Second, ultrametrics satisfy the strong triangle inequality, which is easier to enforce and analyze compared to the more complex four-point condition required for general tree metrics. Finally, ultrametrics are widely used in practice, including hierarchical clustering (Ailon & Charikar, 2011) and phylogenetic tree construction in bioinformatics (Gavryushkin & Drummond, 2016; Yu et al., 2023).

*Table 7.* Comparison of UltraTWD with existing tree-Wasserstein distance methods. The table evaluates four key aspects: (1) the ability to optimize the tree structure, (2) the ability to optimize edge weights, (3) the dependency on multiple trees, (4) the need for embedded data points, and (5) the need for training data. Unlike traditional methods, UltraTWD achieves both tree and edge weight optimization while avoiding the dependency on multiple trees, embedded data points and training data, making it more accurate and robust.

| Method | Tree Structure Optimization | Edge Weight Optimization | No Multiple Trees Needed | No Embedded Data Points Needed | No Training Data Needed |
|---|---|---|---|---|---|
| QuadTree | ✗ | ✗ | ✓ | ✗ | ✓ |
| ClusterTree | ✗ | ✗ | ✓ | ✗ | ✓ |
| qTWD | ✗ | ✓ | ✓ | ✗ | ✓ |
| cTWD | ✗ | ✓ | ✓ | ✗ | ✓ |
| Sliced-QuadTree | ✗ | ✗ | ✗ | ✗ | ✓ |
| Sliced-ClusterTree | ✗ | ✗ | ✗ | ✗ | ✓ |
| Sliced-qTWD | ✗ | ✓ | ✗ | ✗ | ✓ |
| Sliced-cTWD | ✗ | ✓ | ✗ | ✗ | ✓ |
| UltraTree | ✓ | ✓ | ✓ | ✓ | ✗ |
| UltraTWD-MST | ✓ | ✓ | ✓ | ✓ | ✓ |
| UltraTWD-IP | ✓ | ✓ | ✓ | ✓ | ✓ |
| UltraTWD-GD | ✓ | ✓ | ✓ | ✓ | ✓ |

## A.4. Comparison Methods and Limitations

Our approach is evaluated against a range of representative methods designed to approximate 1-Wasserstein distance:

**I. Entropy-Based Method:** The **Sinkhorn** distance (Cuturi, 2013) introduces an entropic regularization term to the objective function to improve the efficiency:

$$W_\lambda(\mu, \nu) = \min_{\gamma \in \Gamma(\mu,\nu)} \sum_{i,j=1}^{n} \gamma_{ij} d(\boldsymbol{x_i}, \boldsymbol{x_j}) - \lambda H(\gamma), \quad (14)$$

where $\lambda > 0$ is the regularization parameter and $H(\gamma)$ is the entropy of the transport plan defined as:

$$H(\gamma) = -\sum_{i,j=1}^{n} \gamma_{ij} \log \gamma_{ij}.$$

The Sinkhorn distance can be efficiently computed using the Sinkhorn-Knopp algorithm with the computational complexity of $\mathcal{O}(n^2)$ (Cuturi, 2013).

**II. Tree-Construction Methods:** They build a hierarchical tree using either QuadTree or ClusterTree structures.

- **QuadTree** (Indyk & Thaper, 2003; Backurs et al., 2020) constructs a hierarchical tree by recursively partitioning the data space, represented as a hypercube in $\mathbb{R}^d$, into smaller sub-hypercubes. The process begins with an initial hypercube, which is iteratively divided into $2^d$ sub-hypercubes. If a sub-hypercube contains exactly one data point, its center is used as a node in the tree. If it contains multiple data points, the partitioning continues until either a predefined depth is reached or no further subdivision is required. Finally, this method constructs a tree by splitting a space into hypercubes.

- **ClusterTree** (Le et al., 2019) adaptively partitions the data space using farthest-point clustering. Unlike fixed

partitioning methods, this algorithm iteratively divides the data into a predefined number of clusters by applying the farthest-point clustering strategy. The centroid of each cluster is designated as a node in the tree, and the partitioning process is recursively repeated for each cluster until the desired depth is reached.

Once the QuadTree or ClusterTree is constructed, the weight $w_e$ of each edge $e$ in the tree is determined by its depth $l(e)$, which represents the number of edges in the unique path from the root to the edge $e$. The edge depth reflects the hierarchical level of the edge, with larger values indicating deeper levels in the hierarchy. Based on this depth, the edge weight is defined as $w_e = 2^{-l(e)}$, leading to the tree-Wasserstein distance formula:

$$W_T(\mu, \nu) = \sum_{e \in T} 2^{-l(e)} \cdot |\mu(T_e) - \nu(T_e)|.$$

**III. Weight-Optimized Methods:** The **qTWD** and **cTWD** methods (Yamada et al., 2022) optimize the edge weights of a fixed QuadTree or ClusterTree structure by solving a Lasso-based regression problem:

$$\min_{\boldsymbol{w} \in \mathbb{R}_+^{n_{\text{node}}}} \sum_{i,j=1}^{n} \left( d(\boldsymbol{x_i}, \boldsymbol{x_j}) - \boldsymbol{w}^\top \boldsymbol{z_{i,j}} \right)^2 + \lambda \|\boldsymbol{w}\|_1, \quad (15)$$

where $\boldsymbol{w} \in \mathbb{R}_+^{n_{\text{node}}}$ represents the vector of edge weights, and $\boldsymbol{z_{i,j}} \in \mathbb{R}_+^{n_{\text{node}}}$ is a vector determined by the tree structure. In this formulation, the tree distance is expressed as $d_T(\boldsymbol{x_i}, \boldsymbol{x_j}) = \boldsymbol{w}^\top \boldsymbol{z_{i,j}}$. The optimization problem aims to adjust the tree distance $d_T$ to align with the cost $d$. The $\ell_1$-regularization term encourages sparsity by driving some edge weights to zero, allowing the corresponding nodes to be merged, thereby reducing the overall tree size.

**IV. Tree-Sliced Methods:** Inspired by the sliced-Wasserstein distance (Rabin et al., 2012), Le et al. (2019) introduced the tree-sliced Wasserstein distance, which includes the **Sliced-QuadTree** and **Sliced-ClusterTree** methods. These methods compute the average tree-Wasserstein distance (TWD) over multiple randomly constructed trees. Similarly, Yamada et al. (2022) proposed sliced versions of qTWD and cTWD, referred to as **Sliced-qTWD** and **Sliced-cTWD**, respectively. The sliced TWD is defined as:

$$W_T(\mu, \nu) := \frac{1}{K} \sum_{i=1}^{K} W_{T_i}(\mu, \nu), \qquad (16)$$

where $W_{T_i}(\mu, \nu)$ represents the TWD computed on the $i$-th tree, and $K$ is the total number of constructed trees.

**V. Supervised Method: UltraTree** (Chen et al., 2024) introduced a supervised approach based on ultrametric trees, where the ultrametric matrix $D_T = [d_T(\boldsymbol{x_i}, \boldsymbol{x_j})] \in \mathbb{R}^{n \times n}$ is optimized by solving a regression problem:

$$\min_{D_T \in \mathbb{R}^{n \times n}} \sum_{(i,j) \in S} \left( W_1^{\text{train}}(\mu_i, \mu_j) - W_T^{\text{train}}(\mu_i, \mu_j) \right)^2, \quad (17)$$

where $S$ denotes the training set, $D_T$ represents the learned ultrametric, $W_1^{\text{train}}$ is the 1-Wasserstein distance pre-computed on the training data, and $W_T^{\text{train}}$ denotes the tree-Wasserstein distance defined on the ultrametric tree $T$. As a supervised approach, it requires extensive training data and the pre-computation of the 1-Wasserstein distance to serve as labels, which can be computationally expensive.

**Limitations of Existing Methods.** The existing Wasserstein approximation methods have several notable limitations, as summarized in Table 7: **(1) The entropy-based method** (Sinkhorn distance) deviates from the true 1-Wasserstein distance due to the entropic regularization term and has a time complexity of $\mathcal{O}(n^2)$. **(2) Tree-construction methods** cannot optimize the tree structure and edge weights and require the embedded data points $X = [\boldsymbol{x_1}, \ldots, \boldsymbol{x_n}]$ to partition the data space. **(3) Weight-optimized methods** also cannot optimize the tree structure and depend on randomly constructed QuadTree or ClusterTree. **(4) Tree-sliced methods** require multiple trees but still lack the capability to optimize tree structures. Additionally, these unsupervised TWD methods, based on QuadTree or ClusterTree, rely on embedded data points. **(5) The supervised method** can optimize tree structures and utilize the distance matrix $D$ as input, eliminating the need for embedded data points. However, they require extensive training data and precomputed 1-Wasserstein distances for training.

In contrast, the proposed UltraTWD methods inherit the advantages of UltraTree and can optimize both the tree structure and edge weights. Moreover, they surpass UltraTree by eliminating the need for training data while offering greater accuracy and efficiency.

**A.5. Other Related Work**

Beyond the comparison methods, trees and hierarchical partitions have been widely explored for approximating the 1-Wasserstein distance. These include an importance sampling method (Indyk, 2007), greedy tree approaches (Khesin et al., 2021; Agarwal et al., 2023; Fox & Lu, 2023; Agarwal et al., 2024), and a streaming algorithm (Chen et al., 2022). However, these works primarily rely on quadtrees, whereas our method focuses on ultrametric trees.

In addition, there are several tree-Wasserstein distances that are not included in our comparison due to different goals and formulations: **(1)** Lin et al. (2025) learns a meaningful tree-Wasserstein distance that captures latent feature hierarchies by using hyperbolic diffusion LCA relations. **(2)** STW (Takezawa et al., 2021) is a supervised method that uses contrastive loss to learn task-specific distances. Since it does not aim to approximate $W_1$, it yields higher errors; for example, on the BBCSport dataset, STW produces a large RE-W of 0.643 and a retrieval precision of 0.335. **(3)** Tran et al. (2024) and Tran et al. (2025) extend the sliced-Wasserstein distance (Rabin et al., 2012) by using structured systems of lines that can be metrized by tree distances. While interpretable as tree-based sliced-Wasserstein variants, they are not typical TWD methods for approximating $W_1$.

In contrast, UltraTWD is specifically designed to accurately approximate the 1-Wasserstein distance by learning both tree structure and edge weights within a unified ultrametric framework.

## B. Experimental Setup

### B.1. Datasets

We evaluate the performance using four benchmark text datasets (Huang et al., 2016): BBCSport, Reuters, Ohsumed, and Recipe. In these datasets, each word is represented as a 300-dimensional word2vec vector, while each document is represented by a normalized bag-of-words distribution based on word frequencies. The details are provided below.

• **BBCSport:** This dataset consists of 737 BBC sports articles labeled by 5 classes. It contains five test sets with 220 articles each, averaging 6,051 words per test set.

• **Reuters:** This dataset contains 7,674 news articles across 8 classes. We randomly generate five test sets with 1,000 articles each, averaging 6,416 words per test set.

• **Ohsumed:** This dataset contains 9,152 medical abstracts within 10 classes. We randomly create five test sets with 1,000 abstracts each, averaging 9,467 words per test set.

• **Recipe:** This dataset consists of 4,370 recipe procedures with 15 classes. It contains five test sets with 1,311 tweets each, averaging 4,084 words per test set.

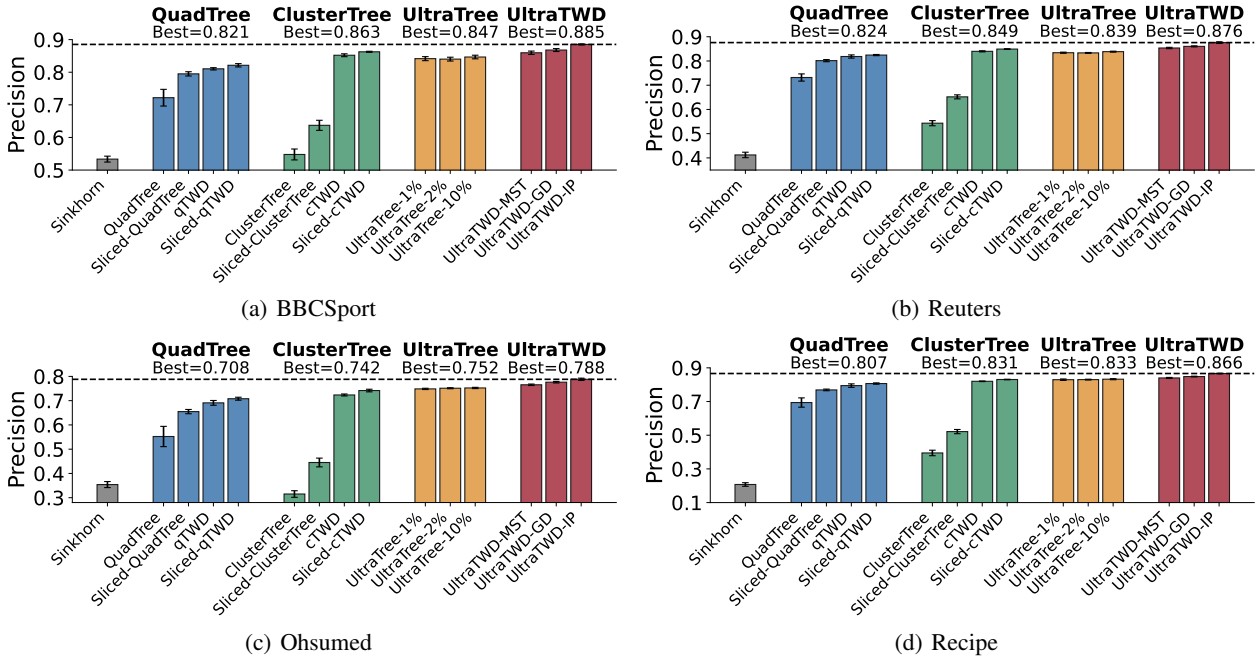

*Figure 6.* Document retrieval precision of tree-Wasserstein distance methods. Blue represents QuadTree-based methods, green represents ClusterTree-based methods, yellow represents UltraTree methods, and red represents UltraTWD methods. Our UltraTWD-IP and GD approaches consistently outperform all baseline methods across four datasets.

## B.2. Implementation and Hyperparameters

The hyperparameters of baseline methods are listed below:

• **Sinkhorn method:** The Sinkhorn distance is computed using the `ot.sinkhorn2` function from the Python Optimal Transport library (Flamary et al., 2021), with a regularization parameter of $\lambda = 1$ in Eq. (14) and a maximum number of iterations `numItermax = 100`.

• **Unsupervised TWD methods:** We implement QuadTree, ClusterTree, qTWD, cTWD, and their sliced versions using the official code[5] provided by Yamada et al. (2022). For qTWD and cTWD, the regularization parameter in Eq. (15) is set to $\lambda = 0.001$, following the configuration in Yamada et al. (2022). For the tree-sliced methods, the number of multiple trees is set to $K = 3$ in Eq. (16), consistent with Yamada et al. (2022).

• **Supervised TWD method:** For UltraTree, five training sets are randomly generated for each dataset, each containing 1,000 document distributions. Each document distribution is created using `numpy.random.randint(1,11,size=(1,n_word))`, with 1% of the entries randomly selected to form a sparse vector, which is then normalized to a probability distribution. The official code[6] is used with default settings: a batch size

---

[5] https://github.com/oist/treeOT
[6] https://github.com/chens5/tree_learning

of 32, the Adam optimizer with a learning rate of 0.01, and a maximum of 5 iterations (typically converging within 2).

## C. Comprehensive Results and Analysis

This section presents a detailed analysis of our comprehensive results, including

• document retrieval performance (Section C.1),
• hyperparameter and convergence analysis (Section C.2).

### C.1. Document Retrieval Performance

We evaluate the average precision for the top-10 retrieved candidates and present the numerical results in Table 3(b). To provide deeper insights, the corresponding visualizations are shown in Fig. 6.

The key observations are as follows:

• **Effect of Optimized Edge Weights:** The qTWD and cTWD methods optimize edge weights based on the QuadTree and ClusterTree structures and thus significantly enhance retrieval performance compared to their non-optimized counterparts (QuadTree & ClusterTree).

• **Impact of Sliced Distances:** The sliced methods improve precision by averaging tree-Wasserstein distances across multiple randomly constructed trees. However, without slicing, the retrieval performance of QuadTree is highly sensitive to tree randomness, leading to greater variance.

• **Stability of UltraTree:** While the sparsity of training data influences the approximation error of $W_T$, UltraTree consistently maintains stable retrieval performance across varying sparsity levels (e.g., 1%, 2%, 10%).

• **UltraTWD Superiority:** Among UltraTWD methods, UltraTWD-IP and GD consistently outperform UltraTWD-MST and all baseline methods. Compared to the best-performing baseline, UltraTWD-IP achieves significant precision gains of 0.022, 0.027, 0.036, and 0.033 on the BBC-Sport, Reuters, Ohsumed, and Recipe datasets, respectively.

The improved precision validates that our UltraTWD methods effectively preserve neighbor relationships, ensuring that documents similar in $W_1$ remain similar in $W_T$.

### C.2. Hyperparameter and Convergence Analysis

**Hyperparameter Analysis.** We analyze the impact of hyperparameters on convergence and retrieval performance.
**(1) UltraTWD-MST:** It directly computes the optimal solution using Theorem 5 and requires no hyperparameters.
**(2) UltraTWD-IP:** The iterative projection method in Algorithm 2 has only one hyperparameter: the maximum number of iterations $m$. As shown in Fig. 7, retrieval precision slightly improves with more iterations on BBCSport, remains stable on Reuters, and decreases slightly on Ohsumed and Recipe datasets. Qualitatively, a decline in performance with additional iterations may indicate overfitting of the $D_T$ solution, weakening $W_T$'s ability to preserve neighbor relationships. Despite these fluctuations, UltraTWD-IP consistently achieves the highest retrieval precision, outperforming UltraTWD-GD and all baseline methods. Considering the time cost of iterative projection, we set $m = 1$ and use results from **a single iteration** in practical applications, as it already delivers excellent performance.
**(3) UltraTWD-GD:** The gradient descent method in Algorithm 3 has two hyperparameters: the learning rate $\alpha$ and the maximum number of iterations $m$. As shown in Fig. 7, $\alpha = 0.02$ achieves faster convergence than $\alpha = 0.01$ and smoother convergence than $\alpha = 0.05$, leading to higher precision than $\alpha = 0.01$ and more stable performance than $\alpha = 0.05$. A large learning rate ($\alpha = 0.05$) may cause overshooting or oscillations around local minima, leading to suboptimal solutions. Empirically, the choice of $\alpha$ depends on the specific dataset, and we use $\alpha = 0.02$ in this paper. For the maximum number of iterations, we set $m = 8$ in practice, balancing low RE-D and high precision.

**Convergence Analysis.** To illustrate the optimization process, we evaluate the relative error of the learned ultrametric $D_T$ from different iterations, defined as

$$\text{RE-D} = \frac{\|D_T - D\|_F}{\|D\|_F},$$

which is proportional to the objective function $\|D_T - D\|_F^2$.

As shown in Fig. 7, both UltraTWD-IP and GD methods exhibit empirical convergence across all four datasets. Regardless of whether iterative projection or gradient descent is employed, the objective function $\|D_T - D\|_F$ is consistently minimized to comparable values. Although global optimality is challenging to guarantee, the $D_T$ obtained using these methods yields high-quality $W_T$, resulting in high retrieval precision. The precision follows the order of IP > GD ($\alpha = 0.02$) > MST > most baselines, validating the effectiveness of our methods.

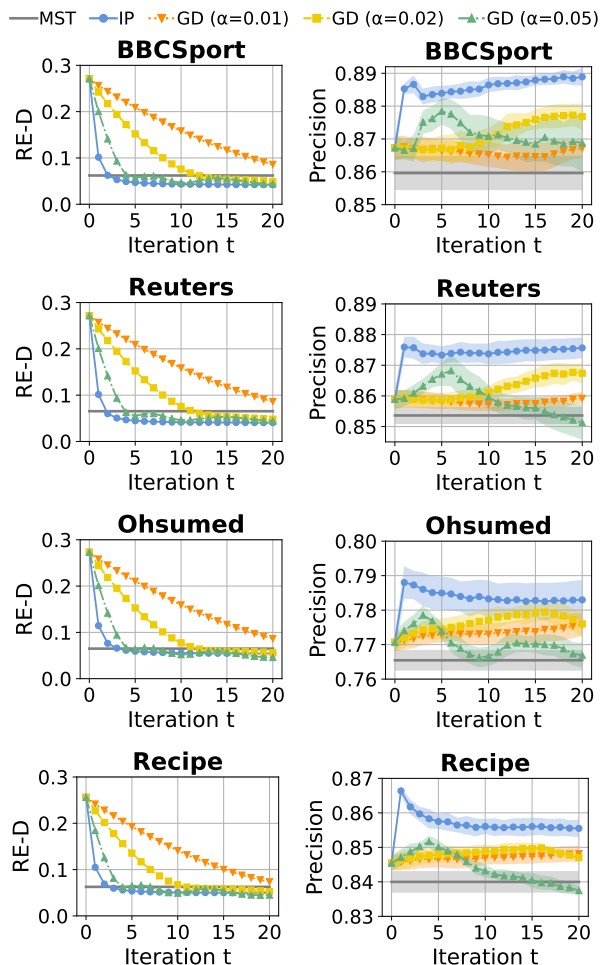

*Figure 7.* Hyperparameter analysis of UltraTWD methods across four datasets, illustrating empirical convergence.

