# OpenReview forum: "UltraTWD: Optimizing Ultrametric Trees for Tree-Wasserstein Distance"
_ICML.cc/2025/Conference — ICML 2025 poster_

### Official Review · Reviewer_XieA · 2025-03-08

**Overall Recommendation:** 3

**Summary:**

This paper introduces an unsupervised approach to constructing an ultrametric tree for the Tree-Wasserstein Distance (TWD) that approximates the Wasserstein distance.

**Claims And Evidence:**

- The method relies on a tree-based representation, yet the paper does not clearly define what constitutes the tree (e.g., whether it is strictly binary or can be flexible) or how the structure is updated.
- The optimization goal is related to metric multidimensional scaling, but there is no discussion about it.
- Although the authors claim an efficient approximation of the Wasserstein distance, the complexity analysis (especially the O(n³) cost for UltraTWD-IP) does not clearly indicate a substantial improvement over the O(n³ log n) complexity of traditional optimal transport methods.

**Essential References Not Discussed:**

- The triplet constraints or triplet relation were explored in previous work, e.g., An Improved Cost Function for Hierarchical Cluster Trees. Although they have not yet been applied within iterative projection frameworks, it would be valuable to reference this earlier research considering the triplet in tree relation.
- Related work not discussed:
    - Distance-Based Tree-Sliced Wasserstein Distance
    - Tree-Wasserstein Distance for High Dimensional Data with a Latent Feature Hierarchy
    - Projection Optimal Transport on Tree-Ordered Lines

**Experimental Designs Or Analyses:**

- Document datasets are appropriate, and the empirical evaluation is comprehensive regarding performance metrics.
- The absence of experiments comparing with supervised tree-Wasserstein distance and WMD limits our understanding of the method’s benefits.
- There is no clear explanation of how the tree structure is updated.

**Methods And Evaluation Criteria:**

The proposed methods appear generally appropriate for document data applications and the use of Wasserstein distance approximations. The experimental evaluation is a strength; however, since the paper posits the paper’s competing work with the supervised method, it would be better to also include the results from the supervised tree-Wasserstein distance by Takezawa et al. and discuss this related work. In addition, since the paper claims to be an efficient approximation of Wasserstein distance and the application is document data, the empirical results of WMD should be included in the comparison.

**Other Comments Or Suggestions:**

NA

**Other Strengths And Weaknesses:**

Strengths:
- The paper is well-written and easy to follow.
- Extensive experimental results on document datasets.

Weaknesses:
- Minor issues include undefined symbols (e.g., $n$ in Section 1, the norm in Eq. (7), and the height in Eq. (8)).

**Questions For Authors:**

- Could you provide a clear definition of the tree structure used in your method? Specifically, is the method limited to binary trees, or does it accommodate more flexible tree structures?
- How is the tree structure updated during the iterative projection process? Does the update involve regrouping leaves or modifying the overall tree topology?
- Can you elaborate on the practical benefits of the O(n³) complexity for UltraTWD-IP, especially when compared to the O(n³ log n) complexity of traditional optimal transport methods?
- When reporting the computation time in Figures 7 and 8 in Appendix C.3, which algorithm was referred to in UltraTWD?
- Could you include empirical comparisons with supervised tree-Wasserstein distance methods and standard WMD, given that your method claims to be an efficient approximation?
- Could you provide a visualization of the constructed tree from the proposed methods?

**Relation To Broader Scientific Literature:**

The paper contributes to the literature on efficient approximations of the Wasserstein distance for document data.

**Theoretical Claims:**

The complexity claim for UltraTWD-IP is stated as O(n³). The improvement over classical optimal transport (O(n³ log n)) is not significant enough to be a clear theoretical advantage.

---

> ### Author Rebuttal · Authors · 2025-04-01
>
> Thank you for your valuable questions. **We will revise the paper accordingly and add more discussion on related work.**
>
> ---
>
> **Question 1.** *Could you provide a clear definition of the tree structure used in your method? Specifically, is the method limited to binary trees, or does it accommodate more flexible tree structures?*
>
> **Answer 1.** Our method is based on ultrametric trees, which are rooted but not necessarily binary. While we empirically followed [1] and used the hierarchical minimum spanning tree procedure to construct a binary ultrametric tree, the method itself is not limited to binary structures. Non-binary ultrametric trees can also be constructed using algorithms like UPGMA. We plan to explore more flexible tree structures in future work.
>
> [1] Chen, S., Tabaghi, P., and Wang, Y. Learning ultrametric trees for optimal transport regression. AAAI, 2024.
>
> ---
>
> **Question 2.** *How is the tree structure updated during the iterative projection process? Does the update involve regrouping leaves or modifying the overall tree topology?*
>
> **Answer 2.** Great question. During the iterative projection process, we update the distance matrix $D_T \in \mathbb{R}^{n \times n}$ to satisfy the strong triangle inequalities. This projection operates directly on the matrix entries and does not explicitly modify the tree structure or regroup leaves. The tree topology is constructed only once, in the final step of Algorithm 2, using a minimum spanning tree algorithm applied to the projected matrix.
>
> ---
>
> **Question 3.** *Could you provide a visualization of the constructed tree from the proposed methods?*
>
> **Answer 3.** Please see the [figure](https://anonymous.4open.science/r/rebuttal_twd/visualization.pdf). We will include it in the revision.
>
> ---
>
> **Question 4.** *Can you elaborate on the practical benefits of the $O(n^3)$ complexity for UltraTWD-IP, especially when compared to the $O(n^3 \log n)$ complexity of traditional optimal transport methods?*
>
> **Answer 4.** We would like to clarify that the $O(n^3)$ complexity of UltraTWD-IP refers to a one-time cost for learning the ultrametric tree. After this step, **each pairwise distance $W_T(\mu, \nu)$ can be computed in just $O(n)$ time**-much faster than the $O(n^3 \log n)$ cost of computing exact $W_1(\mu, \nu)$. As discussed in Section 3.6, the total time complexity for computing $W \in \mathbb{R}^{N \times N}$ is:
> - **$O(N^2 \cdot n + n^3)$** for UltraTWD-IP
> - **$O(N^2 \cdot n + n^2)$** for UltraTWD-GD
> - **$O(N^2 \cdot n^3 \log n)$** for traditional $W_1$.
>
> For example, on the BBCSport dataset with 3,000 valid words per distribution, computing $W_T \in \mathbb{R}^{100 \times 100}$ using UltraTWD-IP takes only **0.9 hours** (including 0.3 hours for tree learning), compared to **2.3 hours** with traditional optimal transport.
>
> ---
>
> **Question 5.** *When reporting the computation time in Figures 7 and 8, which algorithm was referred to in UltraTWD?*
>
> **Answer 5.** Apologies for the confusion. Since all three UltraTWD variants use the same type of ultrametric tree, their computation times are nearly identical. We reported the average time across three algorithms and will clarify this in the revision.
>
> ---
>
> **Question 6.** *Could you include empirical comparisons with supervised tree-Wasserstein distance methods and standard WMD, given that your method claims to be an efficient approximation?*
>
> **Answer 6.** As suggested, we will include the following comparisons:
>
> - **Supervised methods:** We compare with STW [1] and UltraTree [2] (UltraTree is discussed in the main text, Line 367, page 7). STW learns a new distance via contrastive loss rather than approximating $W_1$, resulting in large errors and poor precision. STW training is also time-consuming, taking **3.4 hours** on BBCSport, whereas UltraTWD-GD requires only **34 seconds** to train. As shown in Table R1, our methods consistently outperform both STW and UltraTree in accuracy and efficiency.
>
>   **Table R1. Performance comparison on the BBCSport dataset.**
>   |Metric|RE-W$\downarrow$|Precision$\uparrow$|Total Time (hour)|
>   -|-|-|-
>   STW|0.643|0.335|3.5
>   UltraTree|0.022|0.842|0.6
>   UltraTWD-GD|0.016|0.868|**0.1**
>   UltraTWD-IP|**0.014**|**0.885**|0.5
>
>   [1] Takezawa, Y., Sato, R., & Yamada, M. Supervised tree-wasserstein distance. ICML, 2021.
>
>   [2] Chen, S., Tabaghi, P., and Wang, Y. Learning ultrametric trees for optimal transport regression. AAAI, 2024.
>
> - **WMD:** Standard WMD is used as the ground-truth $W_1$ in the main text. Figures 7 and 8 (page 16) show our methods are significantly faster, even when including tree learning time (see also **Answer 4**). Table R2 compares SVM classification accuracy, where WMD yields the highest accuracy due to exact optimization, while our methods remain competitive with much lower cost.
>
>   **Table R2. Comparison of SVM classification accuracy.**
>   |Dataset|BBCSport|Reuters
>   -|-|-
>   WMD|**0.874**|**0.919**
>   UltraTWD-GD|0.838|0.900
>   UltraTWD-IP|0.839|0.905

---

> > ### Comment · Reviewer_XieA · 2025-04-03
> >
> > I thank the authors for their response. Most of my concerns were addressed. Could you please elaborate on how you plan to incorporate and discuss the related work in your manuscript?

---

> > > ### Author Response · Authors · 2025-04-04
> > >
> > > We're very glad to see that most concerns have been addressed. To provide a comprehensive discussion, we summarize the related works in the following three categories:
> > >
> > > - **Comparison with Triplet-Based Methods:** Both UltraTWD and prior methods [1, 2] leverage triplet information for tree construction, but differ in motivation and formulation. [1] introduces a ratio-cost function based on lowest common ancestor (LCA) relations to recover hierarchical clustering structures. [2] extends this idea by defining a Hyperbolic Diffusion LCA (HD-LCA) to capture latent feature hierarchies, followed by computing a tree-Wasserstein distance on the resulting tree. In contrast, UltraTWD directly optimizes ultrametric trees by enforcing triplet constraints of $d_{ij}^T \le \max ( d_{ik}^T, d_{jk}^T ) $, minimizing $||D_T - D||_F^2$ via iterative optimization. Rather than pursuing hierarchical consistency, UltraTWD focuses on accurately approximating the Wasserstein distance through principled ultrametric learning.
> > >
> > >   [1] Wang, D., & Wang, Y. An improved cost function for hierarchical cluster trees. Journal of Computational Geometry, 2020.
> > >
> > >   [2] Lin, Y. W. E., Coifman, R. R., Mishne, G., & Talmon, R. Tree-Wasserstein distance for high dimensional data with a latent feature hierarchy. ICLR, 2025.
> > >
> > > ---
> > >
> > > - **Comparison with Line-Based Methods:** Recent works [3, 4] approximate Wasserstein distance by projecting measures onto structured line systems, termed tree systems. These methods generalize the Sliced-Wasserstein (SW) distance by replacing single lines with connected line systems metrized by tree distances, making them interpretable as tree-based SW variants. When the system includes only one line, they reduce to standard SW. In contrast, UltraTWD learns a single ultrametric tree that closely approximates the cost matrix, providing a more accurate representation of Wasserstein geometry. As shown in Table R3, UltraTWD achieves significantly lower approximation error than standard SW [5] with 1,000 random projections, making it more suitable for high-precision tasks. *(We may not have time to include [3, 4] in current experiments but will incorporate them in the revision.)*
> > >
> > >   **Table R3. Performance comparison on the BBCSport dataset.**
> > >   |Metric|RE-W$\downarrow$|Precision$\uparrow$|MRR$\uparrow$|ACC$\uparrow$|Total Time (min)
> > >   -|-|-|-|-|-
> > >   SW|0.567|0.466|0.421|0.800|10
> > >   UltraTWD-GD|0.016|0.868|0.921|0.838|28
> > >   UltraTWD-IP|**0.014**|**0.885**|**0.924**|**0.839**|**9**
> > >
> > >   [3] Tran, H. V., Pham, H. T., Huu, T. T., Nguyen-Nhat, M. K., Chu, T., Le, T., & Nguyen, T. M. Projection optimal transport on tree-ordered lines. 2025.
> > >
> > >   [4] Tran, H. V., Nguyen-Nhat, M. K., Pham, H. T., Chu, T., Le, T., & Nguyen, T. M. Distance-based tree-sliced Wasserstein distance. ICLR, 2025.
> > >
> > >   [5] Bonneel, N., Rabin, J., Peyré, G., & Pfister, H. Sliced and radon wasserstein barycenters of measures. Journal of Mathematical Imaging and Vision, 2015.
> > >
> > > ---
> > >
> > > - **Comparison with Supervised Methods:** Both STW [6] and UltraTree [7] are supervised methods for learning tree-based distances but differ in their objectives. STW trains with a contrastive loss to learn task-specific distances without approximating $W_1$, resulting in higher errors and costly training. UltraTree learns to regress tree distances to precomputed Wasserstein distances but requires expensive supervision. In contrast, UltraTWD learns unsupervised ultrametric trees and achieves superior performance with high efficiency. For example, on the BBCSport dataset, STW takes 3.4 hours to train, whereas UltraTWD-GD completes in only 34 seconds.
> > >
> > >   **Table R4. Performance comparison on the BBCSport dataset.**
> > >   |Metric|RE-W$\downarrow$|Precision$\uparrow$|Total Time (hour)|
> > >   -|-|-|-
> > >   STW|0.643|0.335|3.5
> > >   UltraTree|0.022|0.842|0.6
> > >   UltraTWD-GD|0.016|0.868|**0.1**
> > >   UltraTWD-IP|**0.014**|**0.885**|0.5
> > >
> > >   [6] Takezawa, Y., Sato, R., & Yamada, M. Supervised tree-wasserstein distance. ICML, 2021.
> > >
> > >   [7] Chen, S., Tabaghi, P., and Wang, Y. Learning ultrametric trees for optimal transport regression. AAAI, 2024.
> > >
> > > ---
> > >
> > > We will include a dedicated section discussing related works. If space is limited, we will either re-organize the content or move the discussion to the appendix. If any part of the above discussion seems inappropriate or unclear, please feel free to let us know or edit the original review—we will revise accordingly.
> > >
> > > **If you find our response helpful, a higher score would be greatly appreciated. Thank you again for your valuable feedback.**

---

### Official Review · Reviewer_XRdY · 2025-03-10

**Overall Recommendation:** 4

**Summary:**

The Wasserstein distance is a well-known metric for comparing distributions, and has been used as loss function in many ML models. To improve the efficiency of computing the Wasserstein distance, researchers have considered embedding the distributions to a tree metric and computing the Wasserstein distance over the tree embedding, which can be done in linear time. This paper proposes to embed the distributions into a tree that has ultrametric distances, and provide three algorithm for computing the ultrametric tree distances that are close to the underlying metric.

More specifically, the authors formulate the nearness of a tree metric to the ground metric in two ways, namely the $\ell_\infty$ norm and the Frobenius norm of the tree distance and the ground distance. For the $\ell_\infty$ norm, they present a simple algorithm that computes the tree ultrametric using minimum spanning trees. For the Frobenius norm, they present an IPM-based algorithm and a gradient-descent-based approach to compute the ultrametric.

They show in experiments on text datasets that their proposed tree Wasserstein distance (Ultra-TWD) has a lower approximation ratio as well as better performance in document retrieval, ranking, and classification tasks compared to other variants of the tree Wasserstein distance in the literature.


## Update after rebuttal
The authors resolved my concerns. They provided additional experimental results and promised to add the new experimental results as well as the missing citations. I increased my score from 2 to 4.

**Claims And Evidence:**

Yes, most claims are well-supported. I have raised some concerns about theoretical and experimental results below.

**Essential References Not Discussed:**

The use of trees and hierarchical partitionings in approximating the 1-Wasserstein distance has been extensively used in the literature, and I believe the following papers can be cited in this work:

Tree-based Algorithms:
* P. Indyk. "A near linear time constant factor approximation for Euclidean bichromatic matching (cost)." SODA 2007.
* P. K. Agarwal, S. Raghvendra, P. Shirzadian, and R. Sowle. "A higher precision algorithm for computing the 1-Wasserstein distance." ICLR 2023.

MWU-based approaches for boosting the accuracy of greedy tree algorithms:
* A. Khesin, A. Nikolov, and D. Paramonov. "Preconditioning for the Geometric Transportation Problem." SOCG 2019.
* E. Fox and J. Lu. "A deterministic near-linear time approximation scheme for geometric transportation." FOCS 2023.
* P. K. Agarwal, S. Raghvendra, P. Shirzadian, and K. Yao. "Fast and accurate approximations of the optimal transport in semi-discrete and discrete settings." SODA 2024.

Streaming Algorithm:
* X. Chen, R. Jayaram, A. Levi, and E. Waingarten. "New streaming algorithms for high dimensional EMD and MST." STOC 2022.

**Experimental Designs Or Analyses:**

I have a concern with the experiments. Specifically, I am concerned that the experimental setup for competing distances might not be set optimally. For instance, for the Sinkhorn algorithm, the authors set $\lambda=1$ and the maximum number of iterations to 100. Can the authors explain how they chose these parameters, as I expect to see better performance by setting $\lambda=0.01$ and max number of iterations to 300 for instance. I also would like to know how the authors have chosen the regularization parameter $\lambda=0.001$ for the weight-optimized methods and the number 3 of trees in the sliced methods.

**Methods And Evaluation Criteria:**

Yes, the datasets are suitable for the problem.

**Other Comments Or Suggestions:**

-

**Other Strengths And Weaknesses:**

**Strengths**
* The paper is well-written and easy to follow.

**Weaknesses**
* There are no theoretical guarantees on the convergence rate of their proposed IPM and SGD algorithms, and hence, it is hard to judge whether the proposed methods have a comparable time as the previous methods or not.
* The experimental results only compare the proposed methods with the existing TWDs and not other approximations of the Wasserstein distance, such as sliced Wasserstein distance (the SWD and not sliced TWD).

**Questions For Authors:**

-

**Relation To Broader Scientific Literature:**

The problem of computing Wasserstein distance is a well-studied area, and designing fast algorithms that approximate the Wasserstein distnace is of high importance. Approximating the ground distances using a tree metric can help speeding-up the computation of Wasserstein distance, as the Wasserstein distance on a tree metric has a simple closed-form formula that can be computed in linear time.

**Theoretical Claims:**

I have a concern with one of the claims made in Section 3.6: The UltraTWD (or any other tree Wasserstein distances) provides an approximation of the Wasserstein distance. Therefore, comparing the running time of $W_T$ with the exact computation of $W_1$ is unfair. The UltraTWD and an approximation algorithm such as Sinkhorn would have roughly the same execution time (although it is not clear how many iterations does the GD algorithm require).

---

> ### Author Rebuttal · Authors · 2025-03-31
>
> We truly appreciate your positive feedback. Your valuable suggestions will help us improve the work.
>
> ---
>
> **Comment 1.** *Comparing the running time of $W_T$ with the exact computation of $W_1$ is unfair. It is not clear how many iterations does the GD algorithm require.*
>
> **Response 1.** To ensure a fair comparison, we consider the total time complexity required to compute $W \in \mathbb{R}^{N \times N}$.
> - For exact $W_1$: $O(N^2 \cdot n^3 \log n)$.
> - For UltraTWD: (including the tree learning time)
>   - $O(N^2 \cdot n + n^3)$ for UltraTWD-IP
>   - $O(N^2 \cdot n + n^2)$ for UltraTWD-GD
>
> As described in Algorithms 2 and 3, we use **1 iteration** for UltraTWD-IP and **10 iterations** for UltraTWD-GD. On the BBCSport dataset with 3,000 valid words:
> - UltraTWD-GD learns the tree in 34s and computes each $W_T(\mu, \nu)$ in 0.4s using $O(n)$ time,
> - Each $W_1(\mu, \nu)$ takes 1.7s with $O(n^3 \log n)$ time.
>
> To compute $W \in \mathbb{R}^{100 \times 100}$, the total time is **0.9h (IP)** and **0.6h (GD)**, both significantly faster than **2.3h for exact $W_1$**. We will revise this comparison to be more rigorous.
>
> ---
>
> **Comment 2.** *For the Sinkhorn algorithm, the authors set $\lambda=1$ and the maximum number of iterations to 100. Can the authors explain how they chose these parameters? I also would like to know how the authors have chosen the regularization parameter $\lambda=0.001$ for the weight-optimized methods and the number 3 of trees in the sliced methods.*
>
> **Response 2.** We would like to clarify:
> - For Sinkhorn, we followed [1], using $\lambda=1$ and 100 iterations to balance accuracy and runtime. Smaller $\lambda$ and more iterations may improve accuracy but greatly increase computation time.
> - For weight-optimized and sliced methods, we followed [2], using 3 trees. We selected $\lambda=0.001$ based on the highest Pearson correlation with true Wasserstein distance.
>
> As suggested, we test additional $\lambda$ values. As shown in Table R2, larger $\lambda$ improves RE-W but often lowers retrieval performance. UltraTWD methods consistently outperform these baselines across metrics. We will include these results in the revision.
>
> **Table R2. Performance comparison on the BBCSport dataset.**
> |Metric|RE-W$\downarrow$|Precision$\uparrow$|MRR$\uparrow$|ACC$\uparrow$|
> -|-|-|-|-
> Sliced_qTWD (3 trees)
> $\lambda=0.01$|0.142|0.809|0.859|0.821
> $\lambda=0.1$|0.118|0.798|0.852|0.815
> Sliced_cTWD (3 trees)
> $\lambda=0.01$|0.108|0.867|0.898|0.820
> $\lambda=0.1$|0.036|0.823|0.870|0.832
> UltraTWD-GD|0.016|0.868|0.921|0.838
> UltraTWD-IP|**0.014**|**0.885**|**0.924**|**0.839**
>
> [1] Chen, S., Tabaghi, P., and Wang, Y. Learning ultrametric trees for optimal transport regression. AAAI, 2024.
>
> [2] Yamada, M., Takezawa, Y., Sato, R., Bao, H., Kozareva, Z., & Ravi, S. Approximating 1-wasserstein distance with trees. TMLR, 2022.
>
> ---
>
> **Comment 3.** *I believe the following papers can be cited in this work.*
>
> **Response 3.** We will cite these relevant literature in the revision.
>
> ---
>
> **Comment 4.** *There are no theoretical guarantees on the convergence rate of their proposed IP and GD algorithms, and hence, it is hard to judge whether the proposed methods have a comparable time as the previous methods or not.*
>
> **Response 4.** We acknowledge the concern. While formal convergence rate analysis is challenging due to the non-convexity of the problem, our algorithms are designed to be lightweight and stable in practice. As shown in Figure 4 (Section 4.4), both IP and GD exhibit fast empirical convergence within several iterations. Moreover, Table 4 demonstrates that UltraTWD-GD achieves favorable runtime–performance trade-offs, learning trees in just **18–91 seconds** across datasets while outperforming baselines.
>
> ---
>
> **Comment 5.** *The experimental results only compare the proposed methods with the existing TWDs and not other approximations of the Wasserstein distance, such as sliced Wasserstein distance (the SWD and not sliced TWD).*
>
> **Response 5.** As noted in Section 1, *our work focuses on advancing the TWD framework*. While sliced Wasserstein and its variants require equal-sized point clouds and **are not directly applicable** to our bag-of-words distributions, we reformulate the text data into weighted point clouds by treating each word as a point. We then compare with two sliced OT methods that support unequal masses—**SOPT [1] and SPOT [2]**. As shown in Table R3, **our methods achieve significantly better performance across all metrics**.
>
> **Table R3. Performance comparison on the BBCSport dataset.**
> |Metric|RE-W$\downarrow$|Precision$\uparrow$|MRR$\uparrow$|ACC$\uparrow$|
> -|-|-|-|-
> SOPT|113.6|0.082|0.079|0.357
> SPOT|0.666|0.064|0.063|0.231
> UltraTWD-GD|0.016|0.868|0.921|0.838
> UltraTWD-IP|**0.014**|**0.885**|**0.924**|**0.839**
>
> [1] Bai, Y., Schmitzer, B., Thorpe, M., & Kolouri, S. Sliced optimal partial transport. CVPR, 2023.
>
> [2] Bonneel, N., & Coeurjolly, D. Spot: sliced partial optimal transport. ACM TOG, 2019.

---

> > ### Comment · Reviewer_XRdY · 2025-04-02
> >
> > I thank the authors for their thorough response. I would like to follow up on a few of my earlier comments:
> >
> > Comment 1: My concern remains unresolved. My objection is that UltraTWD does not compute the exact Wasserstein distance. Therefore, comparing its running time with that of an exact computation is not particularly informative. A more appropriate comparison would be between UltraTWD and an approximation method such as Sinkhorn. UltraTWD takes $O(N^2 n^2)$ or $O(N^2 n^3)$ time under the assumption that the distributions have disjoint supports, while the Sinkhorn algorithm also runs in $O(N^2 n^2)$ time.
> >
> > Comment 2: Could you please revisit the comparison of running times between Sinkhorn (with smaller $\lambda$ values and a higher number of iterations) and your method? I believe Sinkhorn can remain fast even with significantly more than 100 iterations, and the regularization parameter should not drastically affect the running time. The fact that Sinkhorn yields higher error than Quadtree in your experiments strongly suggests that the parameters may not have been set correctly.
> >
> > Comment 5: It seems the problem formulation or the chosen parameters in your experiments might need adjustment, as an accuracy of 8% is unexpectedly low. For example, you could consider sampling from the bag-of-words distributions and computing the sliced Wasserstein distance directly, rather than using a partial version.

---

> > > ### Author Response · Authors · 2025-04-03
> > >
> > > **Response 1.** Thank you for the follow-up, but we respectfully disagree with the premise of the objection. The family of tree-Wasserstein distance is NOT designed for arbitrary distributions with disjoint supports. Instead, it targets scenarios with a **fixed total support** (e.g., a shared vocabulary of $n_\text{all}$ word embeddings). Thus, **a single tree is learned once over the shared support and reused across all distribution pairs**. This enables each $W_T(\mu, \nu)$ to be computed in $O(n_\text{all})$ time. By contrast, Sinkhorn handles disjoint supports but incur $O(n^2)$ per pair.
> > >
> > > We clarify this with a new experiment: $N=100$ distributions, each with $n$ random words sampled from a fixed vocabulary ($n_\text{all}=1000$). We compare the total runtime to compute $W \in \mathbb{R}^{N\times N}$:
> > > - **Sinkhorn:** $O(N^2 \cdot n^2)$
> > > - **UltraTWD-IP/GD:** $O(N^2 \cdot n_\text{all} + n_\text{all}^3)$ or $O(N^2 \cdot n_\text{all} + n_\text{all}^2)$
> > >
> > >   A single tree is learned once, enabling linear-time computation.
> > > - **UltraTWD-IP/GD-pairwise:** $O(N^2 \cdot (n^3+n))$ or $O(N^2 \cdot (n^2+n))$
> > >
> > >   A separate tree is learned for each distribution pair; **however, this setup is not the intended use case of TWD.**
> > >
> > > **Table R4. Total time (second) comparison.**
> > > |Support size $n$|100|200|500|800|
> > > -|-|-|-|-
> > > Sinkhorn ($\lambda=0.01$, 300 iterations)|54|79|217|492
> > > UltraTWD-IP (learn a tree using 6s)|**8**|**10**|**14**|**17**
> > > UltraTWD-GD (learn a tree using 1s)|**4**|**6**|**10**|**13**
> > > UltraTWD-IP-pairwise|299|1657|12277|25180
> > > UltraTWD-GD-pairwise|562|1261|3518|5212
> > >
> > > **Conclusion:**
> > > - UltraTWD is highly efficient **when a fixed support is available**, computing all pairwise distances in **linear time** per pair.
> > > - Sinkhorn is more flexible for **arbitrary supports**, but incurs significantly higher cost when the support is fixed.
> > > - We acknowledge UltraTWD is not suitable for fully disjoint supports—but that is **not its design goal.**
> > >
> > > ---
> > >
> > > **Response 2.** As suggested, we test the sinkhorn algorithm with smaller $\lambda$ and more iterations. By definition, as $\lambda \to 0$, the Sinkhorn distance converges to the exact $W_1$, at a substantial computational cost. As shown in Table R5, with $\lambda=0.01$ and 300 iterations, Sinkhorn achieves lower RE-W, but its runtime becomes almost twice as long as UltraTWD. This cost will scale poorly as the dataset size grows.
> > >
> > > **UltraTWD is specifically designed for large-scale computation with shared supports**, where it offers:
> > > - **Faster runtime**, especially when the number of distributions $N$ is large.
> > > - **Competitive accuracy** with orders of magnitude smaller complexity.
> > >
> > > We will revise the experimental section to clarify the trade-offs and ensure the Sinkhorn baseline is treated more rigorously.
> > >
> > > **Table R5. Updated comparison on the Recipe dataset.**
> > > |Metric|RE-W|SVM Accuracy|Total Time (hour)|
> > > -|-|-|-
> > > Sinkhorn ($\lambda=1$, 100 iterations)|0.244|0.381|0.1
> > > Sinkhorn ($\lambda=0.01$, 300 iterations)|0.002|0.498|2.2
> > > UltraTWD-IP|0.026|0.495|1.2
> > > UltraTWD-GD|0.023|0.495|1.1
> > >
> > > ---
> > >
> > > **Response 5.** As suggested, we re-evaluated sliced Wasserstein variants. For each document, we sampled 100 points from its bag-of-words distribution. We then followed the official implementations of [1] to test SW [2], MaxSW [3], and KSW [4] using default settings (e.g., 1000 random projections).
> > >
> > > It is important to note that **sliced Wasserstein distances are designed for computational efficiency—not for accurately approximating $W_1$**. As a result, they produce large errors and low retrieval performance, as confirmed in Table R6. Although SW and KSW achieve reasonable SVM accuracy (~0.80), **UltraTWD consistently outperforms them by over 3%**. This is because UltraTWD emphasizes structure-aware accuracy, not just speed. It better captures the true Wasserstein geometry, making it more suitable for high-precision tasks.
> > >
> > > **Table R6. Performance comparison on the BBCSport dataset.**
> > > |Metric|RE-W$\downarrow$|Precision$\uparrow$|MRR$\uparrow$|ACC$\uparrow$|Total Time (min)
> > > -|-|-|-|-|-
> > > SW|0.567|0.466|0.421|0.800|10
> > > MaxSW|21.79|0.163|0.123|0.711|**4**
> > > KSW|0.560|0.469|0.433|0.809|33
> > > UltraTWD-GD|0.016|0.868|0.921|0.838|28
> > > UltraTWD-IP|**0.014**|**0.885**|**0.924**|**0.839**|9
> > >
> > > [1] Markovian sliced Wasserstein distances: Beyond independent projections. NeurIPS, 2023.
> > >
> > > [2] Sliced and radon wasserstein barycenters of measures. Journal of Mathematical Imaging and Vision, 2015.
> > >
> > > [3] Max-sliced wasserstein distance and its use for gans. CVPR, 2019.
> > >
> > > [4] Orthogonal estimation of Wasserstein distances. AISTATS, 2019.
> > >
> > > ---
> > >
> > > **We’ve provided a thorough comparison and will revise accordingly. We hope the reviewer can focus on the core contribution—*tree-Wasserstein distance*—where we demonstrate clear strengths in unsupervised learning, joint optimization, and the balance between accuracy and efficiency. If you find it valuable, a higher score would mean a lot. Thank you for your feedback.**

---

### Official Review · Reviewer_dNo2 · 2025-03-14

**Overall Recommendation:** 2

**Summary:**

1. Wasserstein distance has been applied to many tasks. This paper mainly focuses on how to learn an optimal tree-Wasserstein distance, which is not limited to a specific task.
2. The primary motivation of this paper is to address the suboptimal tree structures and inadequately tuned edge weights in traditional tree-Wasserstein distance. The proposed new framework (UltraTWD) is the first unsupervised framework to simultaneously optimize both tree structure and edge weights by leveraging the ultrametric property.
3. The authors formulate ultrametric nearness problems to optimize trees equipped with the nearest ultrametric to a cost matrix and propose efficient algorithms to address them. These algorithms are based on minimum spanning trees, gradient descent, and an accurate method using iterative projection, all of which deliver high-quality solutions.
4. The proposed new framework achieves the lowest estimation errors compared to both unsupervised methods and the state-of-the-art
supervised method across four benchmark datasets. Additionally, it demonstrates exceptional performance in document retrieval, ranking, and classification, showcasing its practicality for Wasserstein-distance-based applications

**Claims And Evidence:**

Yes. The claims in the submission are supported by clear and convincing evidence, such as the complexity analysis and experiments.

**Essential References Not Discussed:**

No

**Experimental Designs Or Analyses:**

The author's experiment is relatively simple, and I don't think there are any significant problems in these experiments.

**Methods And Evaluation Criteria:**

I think that the proposed methods and/or evaluation criteria (e.g., benchmark datasets) make sense for the problem or application at hand.

**Other Comments Or Suggestions:**

No

**Other Strengths And Weaknesses:**

Strengths:
1. The motivation of this paper is clear. The authors mainly address the limitations of suboptimal tree structures and inadequately tuned edge weights in traditional methods. The studied problem is meaningful.
2. This paper is well organized, logically clear, and written fluently, making it easier for readers to understand.
3. In this paper, the authors address a key problem: How to bridge the gap between WT and W1. A series of algorithms have been proposed to solve this problem. The author calculated the complexity of the method and showed that his method has lower complexity. Combining minimum spanning trees, iterative projection, and gradient descent provides a robust and innovative approach to optimizing tree structures and edge weights.
4. The current experiments are Ok.

Weaknesses:
1. The rationale for the core idea is not fully explained. The authors formulated the tree-metric nearness problem to closely approximate D.  Specifically, the authors use Wasserstein distance as a constraint to control this tree-metric. This idea is good. The authors aim to make the TWD distance as close as possible to a certain distance to improve its accuracy. There is a problem here:  The authors use approximation strategies throughout the paper but do not analyze the difference between this approximation and real Wassestein distance. Is the closer your distance is to the Wasserstein distance, the better the performance of your new distance will be. If so, it stands to reason that the traditional distance effect is better than your method. If not, what is the significance of this approximation? Your experiments have verified that your distance is better than the traditional Wasserstein distance. How to explain this?
2. Theoretical analysis is not enough. 1）Although the authors provide some algorithms to solve the proposed problem, the paper does not provide strong theoretical guarantees on the convergence or optimality of the proposed algorithms. A more rigorous theoretical analysis would enhance the credibility of the framework. 2）How close the proposed distance is to the traditional distance has not been analyzed theoretically. For example, there is no theoretical analysis of the error between your new method and the Wasserstein distance. 3）The generalization analysis of the proposed new method is also lacking.

**Questions For Authors:**

See Weaknesses

**Relation To Broader Scientific Literature:**

Compared with other related literature, the key contribution of this paper is to improve the accuracy of traditional methods. I think this paper has strong application and may be applied to other fields of natural science, such as image analysis, pattern recognition, etc.

**Theoretical Claims:**

All the author's theoretical analysis comes from other references, so I think there is no need to check the correctness of any proofs for theoretical claims.

---

> ### Author Rebuttal · Authors · 2025-03-30
>
> Thank you for your thoughtful review, and we sincerely appreciate your recognition of the strengths of our work, including the clear motivation, innovative algorithmic design, and strong application. We will revise the paper accordingly.
>
> ---
>
> **Comment 1.** *Is the closer your distance is to the Wasserstein distance, the better the performance of your new distance will be. If so, it stands to reason that the traditional distance effect is better than your method. If not, what is the significance of this approximation? Your experiments have verified that your distance is better than the traditional Wasserstein distance. How to explain this?*
>
> **Response 1.** Thank you for raising this important point. we apologize for any confusion and would like to clarify:
>
> In general, the closer a distance is to the true Wasserstein distance, the better it preserves the underlying geometric structure, which is beneficial for downstream tasks. Our method aims to approximate $W_1$ more accurately than existing tree-based approaches, and it outperforms other approximations. However, **it still performs slightly worse than the true Wasserstein distance**. In the main text, we did not include comparisons with the true $W_1$ due to computational cost, but we have now conducted these experiments. As shown in Table R1, the true Wasserstein distance achieves the highest classification accuracy, while our method ranks second. We will include these results in the revised version.
>
> **Table R1. Comparison of document classification accuracy.**
> |Dataset|BBCSport|Reuters|Ohsumed|Recipe|
> -|-|-|-|-
> Best baseline|0.826|0.902|0.423|0.495
> Our distance|0.839|0.905|0.428|0.495
> Tradition distance|**0.874**|**0.919**|**0.498**|**0.499**
>
> ---
>
> **Comment 2.** *Although the authors provide some algorithms to solve the proposed problem, the paper does not provide strong theoretical guarantees on the convergence or optimality of the proposed algorithms.*
>
> **Response 2.** Theoretical convergence guarantees are extremely challenging to establish due to the non-convex and NP-hard nature of the problem. Similar to our work, prior studies [1, 2] employing gradient-based approaches also acknowledge the absence of such guarantees. For instance, [1] states: "*While we provide no theoretical guarantee to find the global optimum...*"
>
> Despite this, our methods demonstrate **empirical convergence**, as shown in Figure 4. Moreover, they are grounded in well-established principles: projection theory for UltraTWD-IP and gradient descent for UltraTWD-GD, both of which exhibit robust performance across multiple datasets.
>
> [1] Chierchia, G. and Perret, B. Ultrametric fitting by gradient descent. NeurIPS, 2019.
>
> [2] Chen, S., Tabaghi, P., and Wang, Y. Learning ultrametric trees for optimal transport regression. AAAI, 2024.
>
> ---
>
> **Comment 3.** *How close the proposed distance is to the traditional distance has not been analyzed theoretically.*
>
> **Response 3.** Thank you for pointing this out. Our analysis focuses on empirical approximation quality, reported via relative error metrics RE-D and RE-W (Table 3). Theoretically, we now clarify that the approximation error of $|W_T(\mu,\nu)-W_1(\mu,\nu)|$ can be bounded by the difference in the cost matrices. Specifically, we have:
> \begin{equation}
> |W_T(\mu, \nu) - W_1(\mu, \nu)| \le 2||D||\_\infty + ||D_T - D||\_\infty \le 2||D||\_\infty + ||D_T - D||\_F
> \end{equation}
> This result shows that the closer our learned ultrametric $D_T$ is to the cost matrix $D$, the closer the corresponding tree-Wasserstein distance is to the 1-Wasserstein distance. Since our optimization directly minimizes $||D_T-D||_\infty$ or $||D_T - D||_F^2$, the bound ensures that $W_T$ serves as a meaningful and controlled approximation to $W_1$. We will include this analysis in the revised version.
>
> ---
>
> **Comment 4.** *The generalization analysis of the proposed new method is also lacking.*
>
> **Response 4.** As an unsupervised method, UltraTWD does not rely on labeled training data and is inherently designed to generalize across domains without retraining. **We provide an empirical generalization analysis in Appendix C.4 (Table 7, page 16).** In this analysis, we build the tree using the BBCSport vocabulary and evaluate its generalization by testing on 100 randomly generated distribution pairs with varying sparsity levels, where each distribution contains $n$ valid words (sparsity = 1 - # valid words / # total words).
>
> As shown in Table R2, both UltraTWD-GD and UltraTWD-IP consistently achieve low approximation errors (measured by RE-W), demonstrating strong generalization performance across different levels of sparsity and values of $n$.
>
> **Table R2. Approximation error (RE-W) under different test sparsity.**
> |# valid words ($n$)|1000|2000|3000|4000|5000|
> -|-|-|-|-|-
> Sparsity|17%|33%|50%|66%|83%
> Best baseline|0.116|0.150|0.167|0.176|0.176
> UltraTWD-GD|0.088|0.111|0.122|0.127|0.125|
> UltraTWD-IP|**0.065**|**0.073**|**0.076**|**0.076**|**0.072**

---

### Official Review · Reviewer_SjiH · 2025-03-15

**Overall Recommendation:** 3

**Summary:**

The paper proposed a method to find out an Ultra tree Wasserstein distance. The method is based on minimizing a distance between trees satisfying certain conditions for trees. Algorithm 2 proposed to find the solution under projections, meanwhile Algorithm 3 tries to reduce the computation when avoiding working with pairwise distance between leafs, instead it works with the node heights. The second last section is devoted for testing the proposed method on several datasets.

##update after rebuttal: I keep my score, since the authors mostly explained the difficulties of the problem.

**Claims And Evidence:**

New framework for tree-Wasserstein distance exploit the ultra metric property.

The problem is formulated into ultra metric optimization  questions and proposed algorithm to solve it

Empirical evidences are shown to demonstrate the effectiveness of the methods.

No guarantee for the convergence of the proposed algorithms.

**Essential References Not Discussed:**

No

**Experimental Designs Or Analyses:**

According to Table 2, the UltraTWD-IP and UltraTWD-GP are competitive to each other, no dominates other in metrics RE-W and Precision. Is there any explanation for that, since UltraTWD-IP appears to be more natural and take longer time to optimize. In the same table, UltraTree also produce quite competitive results, does this mean that the proposed method not really improve the overall performance significantly enough?

**Methods And Evaluation Criteria:**

The proposed methods sound reasonable. The metrics are fine.

**Other Comments Or Suggestions:**

Why in Algorithm 2, line 3: the weights are set to be $\frac{1}{t+1}$ and $\frac{t}{t+1}$ in which $t$ is the $t$th iteration?

**Other Strengths And Weaknesses:**

No

**Questions For Authors:**

Please read the above comments

**Relation To Broader Scientific Literature:**

Yeap, I think it is related to all other topics involving tree-data.

**Theoretical Claims:**

All theoretical results are standard in theory of tree Wasserstein distance. The authors do not propose any new theoretical result.
Algorithm 2 would be costly because the number of projections is too large.

For the algorithm 3, theoretically is the $D_T$ determined uniquely by $H_T$? Since the number of entries reduce significantly?

---

> ### Author Rebuttal · Authors · 2025-03-30
>
> Thanks for your valuable comments. We will modify it accordingly.
>
> ---
>
> **Comment 1.** *No guarantee for the convergence of the proposed algorithms.*
>
> **Response 1.** Theoretical convergence is extremely hard to guarantee due to the non-convex and NP-hard nature of the problem. Prior work [1, 2] using similar gradient-based methods also lacks convergence guarantees. As [1] states: "*While we provide no theoretical guarantee to find the global optimum...*". Despite this, our methods show **empirical convergence** (Figure 4), and are built on solid principles: projection theory for UltraTWD-IP and gradient descent for UltraTWD-GD, both demonstrating strong performance across datasets.
>
> [1] Chierchia, G. and Perret, B. Ultrametric fitting by gradient descent. NeurIPS, 2019.
>
> [2] Chen, S., Tabaghi, P., and Wang, Y. Learning ultrametric trees for optimal transport regression. AAAI, 2024.
>
> ---
>
> **Comment 2.** *Algorithm 2 would be costly because the number of projections is too large.*
>
> **Response 2.** Although Algorithm 2 involves $O(n^3)$ projections, each is a **closed-form, constant-time operation**, and we use **only one iteration** in practice (Section 3.4). This keeps it practical even for moderately large datasets. For better scalability, UltraTWD-GD (Algorithm 3) has lower $O(n^2)$ complexity per iteration and offers a strong trade-off between speed and accuracy (Table 4, Section 4.4).
>
> ---
>
> **Comment 3.** *For the algorithm 3, theoretically is the $D_T$ determined uniquely by $H_T$? Since the number of entries reduce significantly?*
>
> **Response 3.** Yes. Given the rooted tree structure $T$, the node height vector $H_T \in \mathbb{R}^{2n-1}$ uniquely determines $D_T$, where $d_T(i, j) = h(\text{LCA}(l_i, l_j))$ is the height of the least common ancestor (LCA) of leaves $l_i$ and $l_j$ (Equation 8). Since the LCA structure is fixed by $T$, $H_T$ fully specifies $D_T$. This reduces the parameter space from $O(n^2)$ in $D_T$ to $O(n)$ in $H_T$, and allows efficient optimization over $H_T$ in Algorithm 3.
>
> ---
>
> **Comment 4.** *According to Table 2, the UltraTWD-IP and GD are competitive to each other, no dominates other in metrics RE-W and Precision. Is there any explanation for that, since UltraTWD-IP appears to be more natural and take longer time to optimize.*
>
> **Response 4.** UltraTWD-IP and GD solve the **same problem** but use different strategies: UltraTWD-IP uses local projections to enforce ultrametric constraints, while UltraTWD-GD applies gradient descent on node heights. Thus, their results may vary slightly. As a result, UltraTWD-IP tends to yield slightly better downstream performance due to precise constraint enforcement, while UltraTWD-GD runs significantly faster. **This trade-off highlights the flexibility of our framework—both methods consistently outperform baselines.**
>
> ---
>
> **Comment 5.** *UltraTree also produce quite competitive results, does this mean that the proposed method not really improve the overall performance significantly enough?*
>
> **Response 5.** While UltraTree is a strong supervised baseline, our UltraTWD methods show clear advantages in performance, generalizability, and efficiency.
>
> - **Performance gap:** UltraTWD-IP and GD consistently outperform UltraTree across tasks. In document retrieval (Precision), UltraTWD-IP significantly improves over UltraTree by **4%–5%** on all datasets (see Table R1).
>
>   **Table R1. Precision comparison of document retrieval.**
>   |Dataset|BBCSport|Reuters|Ohsumed|Recipe|
>   -|-|-|-|-
>   Best Unsupervised|0.863|0.849|0.742|0.831
>   UltraTree|0.842|0.834|0.749|0.830
>   UltraTWD-GD|0.868|0.860|0.776|0.848
>   UltraTWD-IP|**0.885**|**0.876**|**0.788**|**0.866**
>   Improvement $\uparrow$|+5.1%|+5.0%|+5.2%|+4.3%
>
> - **Limitations of UltraTree:** UltraTree requires training data with precomputed Wasserstein distances, which is costly and dataset-specific. Its performance also drops when training data sparsity differs from the test set (Table 3, Section 4.2).
>
> - **Advantages of UltraTWD:** Our methods are fully **unsupervised**, robust to sparsity, and optimize $||D_T - D||_F^2$ directly. UltraTWD-GD achieves better performance while being **up to 36$\times$ faster** than UltraTree in tree learning time (Table 4, Section 4.4), making it much more scalable.
>
> **In summary, UltraTWD methods are more accurate, efficient, and broadly applicable than UltraTree.**
>
> ---
>
> **Comment 6.** *Why in Algorithm 2: the weights are set to be $\frac{1}{t+1}$ and $\frac{t}{t+1}$ in which $t$ is the $t$-th iteration?*
>
> **Response 6.** The weight update in Algorithm 2 follows the HLWB projection scheme (Theorem 6, Section 3.4), where $\sigma_t = \frac{1}{t+1}$ forms a **steering sequence**. This gradually reduces the influence of the initial matrix $D$ while stabilizing updates ($\sigma_t \to 0$ as $t \to \infty$). While simple, this setting ensures convergence in convex settings and shows stable behavior empirically in our non-convex case (Figure 4, Section 4.4).

---

### Decision · Program_Chairs · 2025-05-01

**Decision:**

Accept (poster)

**Comment:**

This paper proposes a new approximation algorithm for the tree-Wasserstein distance. More specifically, the paper proposes a computationally efficient algorithm to approximate a distance by tree-metric and use it for tree-Wasserstein distance. The key difference from the existing method is that the proposed method optimizes both the tree structure and its edge weight. Through experiments, it demonstrates that the proposed approximation method outperforms existing approaches.

The proposed approach is interesting and practically useful. Some reviewers have pointed out that the paper lacks novel theoretical results. However, the proposed method is highly useful, and its effectiveness is well validated by experiments. Most reviewers support the paper, and I also support its acceptance.